# Instantons, renormalons and the theta angle in integrable sigma models

**Marcos Mariño[1⋆], Ramon Miravitllas[1†] and Tomás Reis[1,2‡]**

**1** Département de Physique Théorique et Section de Mathématiques,
Université de Genève, Genève, CH-1211 Switzerland
**2** SISSA, 34136 Trieste, Italy and INFN, Sezione di Trieste, 34127 Trieste, Italy

⋆ Marcos.Marino@unige.ch , † Ramon.MiravitllasMas@unige.ch , ‡ treis@sissa.it

## Abstract

Some sigma models which admit a theta angle are integrable at both $\vartheta = 0$ and $\vartheta = \pi$. This includes the well-known $O(3)$ sigma model and two families of coset sigma models studied by Fendley. We consider the ground state energy of these models in the presence of a magnetic field, which can be computed with the Bethe Ansatz. We obtain explicit results for its non-perturbative corrections and we study the effect of the theta angle on them. We show that imaginary, exponentially small corrections due to renormalons remain unchanged, while instanton corrections change sign, as expected. We find in addition corrections due to renormalons which also change sign as we turn on the theta angle. Based on these results we present an explicit non-perturbative formula for the topological susceptibility of the $O(3)$ sigma model in the presence of a magnetic field, in the weak coupling limit.

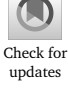

# 1 Introduction

Perturbative series in quantum field theory have typically non-perturbative corrections which are exponentially small in the coupling constant. We expect these corrections to be due either to instantons or to renormalons. Instanton corrections come from non-trivial saddle-points of the path integral and they are in principle accessible with semiclassical methods. In practice, however, instanton calculus is plagued with all kinds of difficulties, the most important one being perhaps the infrared (IR) divergences appearing in asymptotically free theories. Renormalons are even more puzzling, since they have no known description in terms of saddle points of the path integral, and their current understanding relies heavily on estimates from perturbation theory at large order.

One possible avenue for a better understanding of non-perturbative corrections is to look at solvable models where one can extract them from the exact solution. The results obtained in this way give a signpost to test non-perturbative methods. One example of this strategy is the Seiberg–Witten solution of $\mathcal{N} = 2$ supersymmetric Yang–Mills theory [1], which provides a prediction for multi-instanton contributions in this theory. This prediction was first addressed by conventional methods (see [2] for a review), and the effort to understand instantons in this theory from first principles culminated in Nekrasov's work [3]. However, this is a supersymmetric success story in which many generic aspects of non-perturbative effects can not be detected. In particular, $\mathcal{N} = 2$ super Yang–Mills theory has no renormalons, and instanton corrections are one-loop exact.

It has been known for a long time that certain two-dimensional, asymptotically free theories are integrable, and their exact $S$-matrices have been conjectured. These include the $O(N)$ supersymmetric sigma model, the Gross–Neveu model, and the principal chiral field (PCF). These models are expected to have renormalons, and some of them admit instantons and other classical solutions. Therefore, they provide a rich laboratory where one might obtain explicit results for non-perturbative effects, like instanton and renormalon corrections. This issue has been addressed in previous works, by working at large $N$ [4–6] or by using numerical techniques [7–11].[1] In our previous paper [19] we have developed a general method to obtain *analytic, exact* results at finite $N$ for exponentially small corrections to the free energy in these integrable models.[2] Some of these corrections were identified with renormalon effects, and we showed in particular that their structure is more general than the one put forward in the pioneering studies of Parisi [21, 22] and 't Hooft [23]. However, it was pointed out

---

[1]There have been important works on non-perturbative corrections in two-dimensional quantum field theories which do not rely on integrability, but only on large $N$ solvability. These include David's pioneering work on renormalons [12,13], as well as [14–16]. Another, more recent line of work considers twisted compactifications of two-dimensional models, see e.g. [17,18]. However, the calculation of perturbative series and non-perturbative effects in [17,18] and follow-up papers is based on an uncontrolled truncation to an effective quantum mechanical model, and provides at best approximate results.

[2]The recent work [20] obtains analytic results for non-perturbative effects in the $O(4)$ sigma model.

in [19] that some of the corrections found in the $O(N)$ sigma model and in the PCF seem to correspond to instantons. In particular, as emphasized in [24], the leading, real exponentially small correction in the $O(3)$ non-linear sigma model might be due to the well-known instanton configurations in that theory [25].

One important and genuine non-perturbative effect in quantum field theory is the dependence on the topological theta angle, which is completely invisible in perturbation theory. In some cases this dependence is due to instantons, although large $N$ considerations [26] and explicit lattice calculations [27] in pure Yang–Mills theory show that the physics of the theta angle can not be always explained by semiclassical methods. Some integrable two-dimensional models, like the $O(3)$ sigma model, admit also a theta angle. Although turning on a generic theta angle is believed to break integrability, it was observed in [28, 29] that the $O(3)$ sigma model with $\vartheta = \pi$ is still integrable. This picture was generalized by Fendley [30] to two families of coset sigma models: the $SU(N)/SO(N)$ models, and the $O(2P)/O(P) \times O(P)$ models. In these theories there is a $\mathbb{Z}_2$ instanton number, and the theta angle can only take the values $\vartheta = 0, \pi$. Fendley conjectured that these models are also integrable for both values of $\vartheta$, and proposed exact expressions for their $S$-matrices.

In view of the integrability of all these models, one could try to use their exact solutions at $\vartheta = 0, \pi$ to study the behavior of non-perturbative effects as we change the theta angle. This issue was already addressed in [30, 31] by considering deformed versions of these models (in the case of the $O(3)$ sigma model, this deformation is known as the "sausage" model). The calculation of non-perturbative effects in these deformed models is much simpler than in the original ones, and this was part of the motivation to introduce the deformation in the first place. However, the deformed models have various drawbacks: their spacetime physics is not transparent, and they do not seem to have renormalon effects, since the free energy has no perturbative expansion. Instead, all the terms one finds are real exponential corrections. In addition, the original models are recovered from the deformed ones in a singular limit. This means in particular that the free energy of the original, undeformed models cannot be obtained from the free energy of the deformed models calculated in [30, 31]. For this reason, the results of [30, 31] do not give direct access to non-perturbative corrections in the conventional sigma models.

In this paper we study both the $O(3)$ sigma model and Fendley's integrable sigma models, and we consider in detail the effect of the theta angle on their non-perturbative corrections, by using the methods and ideas of [19]. We find that these corrections might change their sign as we move from the theory with $\vartheta = 0$ to the theory $\vartheta = \pi$. Non-perturbative corrections due to instantons change sign according to the value of their topological charge, as expected. In particular, the leading, real exponentially small correction of the $O(3)$ sigma model changes sign as we turn on $\vartheta = \pi$, in agreement with the instanton interpretation put forward in [19, 24]. There are corrections which are due to renormalon effects, and their imaginary part can be detected by a resurgent analysis of the perturbative series. As expected, and required by consistency, this imaginary part remains unchanged as we turn on the theta angle. However, we find that the real part of renormalon corrections *does* change sign and is therefore sensitive to the theta angle.

Our results have also some bearing on long-standing issues concerning the $O(3)$ sigma model. It has been known for some time that the $\vartheta$ dependence of the vacuum energy and the conventional topological susceptibility are not good observables in this model. This is best established by lattice calculations that show that these quantities diverge in the continuum limit (see e.g. [32, 33] for a summary and references). An evaluation directly in the continuum with instanton methods runs into similar problems: the integration over the instanton size is afflicted with the usual IR divergence and an additional logarithmic UV divergence [34–37] (the latter is sometimes regarded as a counterpart of the divergence found in the lattice). Our

results indicate that, in the presence of an $h$ field, the vacuum energy as a function of $\vartheta$ is well defined (after subtracting the result for $h = 0$). The results of this paper and [19] lead to a very precise prediction for the weak-coupling limit of the resulting $h$-dependent topological susceptibility: it is given by the one-instanton effect calculated in [19], see (102) for the explicit result. It might be possible to test this prediction with a lattice calculation.

This paper is organized as follows. In section 2 we consider the $O(3)$ sigma model at $\vartheta = 0$, and we review and extend results of [19] on its non-perturbative corrections. In section 3 we consider the $O(3)$ sigma model at $\vartheta = \pi$, and we analyze the effect of the theta angle on non-perturbative effects. We check our analytic results with a detailed numerical study of the Bethe Ansatz equations for the massless theory at $\vartheta = \pi$. In section 4 we extend these results to Fendley's integrable sigma models. In 5 we list some conclusions and prospects for future research. The first appendix gives some of the details of the computations in section 2, while the second one explains our numerical study of the integral equation when $\vartheta = \pi$.

## 2 The $O(3)$ sigma model at $\vartheta = 0$

### 2.1 The free energy

The observable that we will focus on in this paper is the free energy of integrable sigma models in the presence of a magnetic field or chemical potential $h$. The main advantage of this free energy is that, on the one hand, it can be computed in a simple way by using the Bethe Ansatz [38, 39], and on the other hand it can be studied in perturbation theory in the regime $h \gg \Lambda$, where $\Lambda$ is the dynamically generated scale (in this paper we will always use the $\overline{\text{MS}}$ scheme). For this reason, this quantity has been studied extensively in the past, and famously led to a determination of the ratio $m/\Lambda$, first in the $O(N)$ sigma model [40, 41], and subsequently in many other models [42–47] (see [48] for a review of these results).

The free energy is defined and calculated as follows. Let H be the Hamiltonian of the model, and let Q be the charge associated to a global conserved current. Let $h$ be an external field coupled to Q, which can be regarded as a chemical potential. Let us consider the ensemble defined by the operator

$$\mathsf{H} - h\mathsf{Q}, \tag{1}$$

as well as the corresponding free energy per unit volume

$$F(h) = -\lim_{V,\beta \to \infty} \frac{1}{V\beta} \log \mathrm{Tr}\, \mathrm{e}^{-\beta(\mathsf{H}-h\mathsf{Q})}, \tag{2}$$

where $V$ is the volume of space and $\beta$ is the total length of Euclidean time. The quantity

$$\mathcal{F}(h) = F(h) - F(0), \tag{3}$$

can be computed by using the exact $S$ matrix and the Bethe Ansatz [38, 39]. The solution is encoded in the following integral equation for a Fermi density $\epsilon(\theta)$:

$$\epsilon(\theta) - \int_{-B}^{B} K(\theta - \theta')\epsilon(\theta')\mathrm{d}\theta' = h - m\cosh(\theta), \quad \theta \in [-B, B]. \tag{4}$$

Here, $m$ is the mass of the charged particles, and with a clever choice of Q, it is directly related to the mass gap of the theory. The kernel of the integral equation is given by

$$K(\theta) = \frac{1}{2\pi\mathrm{i}} \frac{\mathrm{d}}{\mathrm{d}\theta} \log S(\theta), \tag{5}$$

where $S(\theta)$ is the appropriate $S$-matrix for the scattering of the particles charged under $Q$. The endpoints $\pm B$ are fixed by the boundary condition

$$\epsilon(\pm B) = 0\,. \tag{6}$$

The free energy is then given by

$$\mathcal{F}(h) = -\frac{m}{2\pi}\int_{-B}^{B}\epsilon(\theta)\cosh(\theta)\mathrm{d}\theta\,. \tag{7}$$

It will also be convenient to use a "canonical" formalism and introduce the density of particles $\rho$ and energy density $e$ through a Legendre transform of $\mathcal{F}(h)$,

$$\rho = -\mathcal{F}'(h)\,,$$
$$e(\rho) - \rho h = \mathcal{F}(h)\,. \tag{8}$$

Alternatively, these two quantities can be obtained from the density of Bethe roots $\chi(\theta)$. This density is supported on an interval $[-B, B]$ and satisfies the integral equation

$$\chi(\theta) - \int_{-B}^{B}K(\theta - \theta')\chi(\theta')\mathrm{d}\theta' = m\cosh\theta\,, \tag{9}$$

where the kernel is the same one appearing in (4). Then, we have

$$e = \frac{m}{2\pi}\int_{-B}^{B}\chi(\theta)\cosh(\theta)\mathrm{d}\theta\,, \qquad \rho = \frac{1}{2\pi}\int_{-B}^{B}\chi(\theta)\mathrm{d}\theta\,. \tag{10}$$

In the canonical formulation, $B$ is fixed by the value of $\rho$, and this leads implicitly to a function $e(\rho)$.

A powerful approach to solve the integral equation (4) is the Wiener–Hopf method, as explained in [40, 43, 49]. We consider the Fourier transform of the kernel,

$$\tilde{K}(\omega) = \int_{\mathbb{R}}\mathrm{e}^{\mathrm{i}\omega\theta}K(\theta)\mathrm{d}\theta\,, \tag{11}$$

and its Wiener–Hopf factorization

$$1 - \tilde{K}(\omega) = \frac{1}{G_+(\omega)G_-(\omega)}\,, \tag{12}$$

where $G_\pm(\omega)$ is analytic in the upper (respectively, lower) complex half plane. Since $K(\theta)$ is an even function, we have $G_-(\omega) = G_+(-\omega)$. We start with the result of the Wiener–Hopf analysis. We define

$$\epsilon_\pm(\omega) = \mathrm{e}^{\pm\mathrm{i}B\omega}\tilde{\epsilon}(\omega)\,, \tag{13}$$

where $\tilde{\epsilon}(\omega)$ is the Fourier transform of $\epsilon(\theta)$, after extending the latter by zero outside the interval $[-B, B]$. Consider also the function

$$g_+(\omega) = \mathrm{i}h\frac{1 - \mathrm{e}^{2\mathrm{i}B\omega}}{\omega} + \frac{\mathrm{i}m\mathrm{e}^{B}}{2}\left(\frac{\mathrm{e}^{2\mathrm{i}B\omega}}{\omega - \mathrm{i}} - \frac{1}{\omega + \mathrm{i}}\right)\,, \tag{14}$$

which is obtained from the Fourier transform of $h - m\cosh(\theta)$, after extending it conveniently outside of $[-B, B]$, as explained in [19]. Then, one finds that (4) is equivalent to the following equation for a function $Q(\omega)$,

$$Q(\omega) - \frac{1}{2\pi\mathrm{i}}\int_{\mathbb{R}}\frac{\mathrm{e}^{2\mathrm{i}B\omega'}\sigma(\omega')Q(\omega')}{\omega + \omega' + \mathrm{i}0}\mathrm{d}\omega' = \frac{1}{2\pi\mathrm{i}}\int_{\mathbb{R}}\frac{G_-(\omega')g_+(\omega')}{\omega + \omega' + \mathrm{i}0}\mathrm{d}\omega'\,, \tag{15}$$

where

$$\sigma(\omega) = \frac{G_-(\omega)}{G_+(\omega)}. \tag{16}$$

The solution $Q(\omega)$ to (15) determines $\epsilon_+(\omega)$ through

$$\frac{\epsilon_+(\omega)}{G_+(\omega)} = \frac{1}{2\pi i}\int_{\mathbb{R}} \frac{G_-(\omega')g_+(\omega')}{\omega' - \omega - i0}d\omega' + \frac{1}{2\pi i}\int_{\mathbb{R}} \frac{e^{2iB\omega'}\sigma(\omega')Q(\omega')}{\omega' - \omega - i0}d\omega'. \tag{17}$$

The relationship between $h$, $m$ and $B$ is determined by the boundary condition (6), which in Fourier space takes the form

$$\lim_{\kappa \to +\infty} \kappa \epsilon_+(i\kappa) = 0. \tag{18}$$

We then find from (7) that

$$\mathcal{F}(h) = -\frac{1}{2\pi}me^B\epsilon_+(i). \tag{19}$$

These equations were analyzed in [40, 41] and subsequent works to obtain the perturbative expansion of $\mathcal{F}(h)$ in the regime $h \gg m$. In the case of "bosonic models", which includes coset sigma models and their supersymmetric extensions, one can obtain a universal formula for the one-loop free energy. In these bosonic models, the Wiener–Hopf factor $G_+(i\xi)$ has the following expansion around $\xi = 0$,

$$G_+(i\xi) = \frac{k}{\sqrt{\xi}}\left(1 - a\xi\log\xi - b\xi + \mathcal{O}(\xi^2)\right), \tag{20}$$

and one finds [42],

$$\mathcal{F}(h) = -\frac{k^2h^2}{4}\left\{\log\left(\frac{h}{m}\right) + \left(a + \frac{1}{2}\right)\log\log\left(\frac{h}{m}\right) + \log\left(\frac{k\sqrt{2\pi}}{G_+(i)}\right)\right.$$
$$\left. + a\left(\gamma_E - 1 + \log(8)\right) - b - 1 + \mathcal{O}\left(\log^{-1/2}(h/m)\right)\right\}. \tag{21}$$

A fully analytic derivation of this formula was presented in [19].

In the case of the $O(3)$ sigma model, we make the choice of charges in [39–41]. The resulting Euclidean Lagrangian, including the $h$ term, is given by

$$\mathcal{L}_h = \frac{1}{2g_0^2}\left\{\partial_\mu \boldsymbol{S}\cdot\partial^\mu \boldsymbol{S} + 2ih(S_1\partial_0 S_2 - S_2\partial_0 S_1) + h^2\left(S_3^2 - 1\right)\right\}, \tag{22}$$

where $\boldsymbol{S} = (S_1, S_2, S_3)$ is the quantum field satisfying the constraint $\boldsymbol{S}^2 = 1$, and $g_0^2$ is the bare coupling constant. This model is asymptotically free [50]. In the convention in which the beta function is given by

$$\beta(g) = \mu\frac{dg^2}{d\mu} = -\beta_0 g^4 - \beta_1 g^6 - \cdots, \tag{23}$$

one has (see e.g. [51])

$$\beta_0 = \frac{1}{2\pi}, \qquad \beta_1 = \frac{1}{4\pi^2}. \tag{24}$$

In the $O(3)$ sigma model, the integral equation (4) has the kernel

$$K(\theta) = \frac{1}{\pi^2 + \theta^2}, \qquad \tilde{K}(\omega) = e^{-\pi|\omega|}, \tag{25}$$

and its Wiener–Hopf decomposition is given by

$$G_+(\omega) = \frac{e^{i\omega\left[-\frac{1}{2}+\frac{\log(2)}{2}\right]+\frac{1}{2}i\omega\log(-i\omega)}}{\sqrt{-i\omega}}\frac{\Gamma(1-i\omega)}{\Gamma\left(\frac{1}{2}-\frac{1}{2}i\omega\right)}. \tag{26}$$

The mass $m$ appearing in (4) is related to the dynamically generated scale by [40]

$$\frac{m}{\Lambda} = \frac{8}{e}, \tag{27}$$

and we note that

$$\Lambda \approx \mu\, e^{-2\pi/g^2(\mu)}, \tag{28}$$

where $g^2(\mu)$ is the renormalized coupling at the scale $\mu$.

## 2.2 Non-perturbative corrections

Since $\mathcal{F}(h)$ is an observable determined by a simple integral equation, it provides an ideal testing ground to determine both the perturbative expansion and its non-perturbative corrections. This was first explored in the case of the principal chiral field, and in the large $N$ limit, in [4,5]. One possible strategy to calculate exponentially small corrections is to use the theory of resurgence combined with perturbation theory at large orders. Volin [7,8] developed a systematic procedure to extract long perturbative series for $\mathcal{F}(h)$ directly from the Bethe Ansatz equations, and this was used in [6–11,20,52] to study numerically and analytically the presence of renormalon effects.

In this paper we will study non-perturbative effects in various integrable sigma models by using the method developed in [19], which provides exact, analytic results for the exponentially small corrections to the free energy, at finite $N$. The first non-perturbative correction in the $O(3)$ sigma model was obtained in [19] at the very first orders in the coupling constant. The calculation in [19] was verified and extended to NNLO in [24].

In order to incorporate non-perturbative corrections in the Wiener–Hopf method, we deform the integration contour along the real axis in (15) into a Hankel contour $\mathcal{C}$ around the positive imaginary axis. This contour is made of two rays, one of them to the left of the

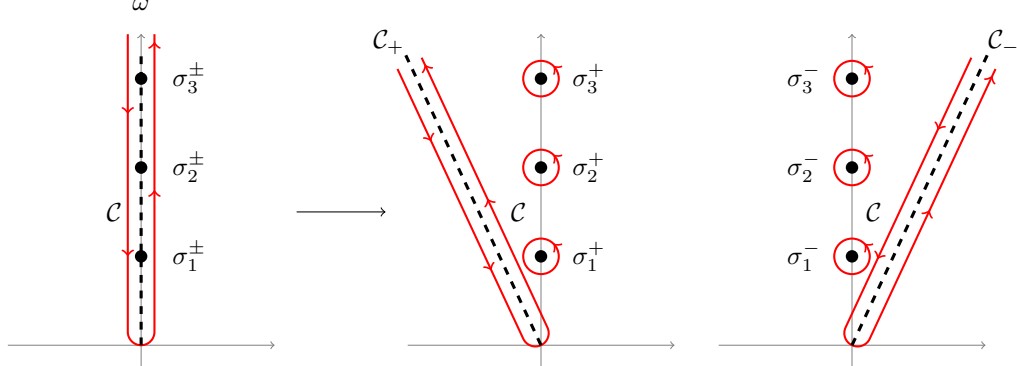

Figure 1: The Hankel contour $\mathcal{C}$ can be deformed into an integral along the discontinuity of $\sigma(\omega)$, denoted by the dashed line, plus a sum over residues. However, due to the branch cut along the imaginary axis, this can be done in two different ways, which leads to two different integrations along the discontinuity, corresponding to the contours $\mathcal{C}_\pm$. The residues of the poles $\sigma_n^\pm$ will also depend on this choice.

imaginary axis, and the other one to the right. If $\sigma(\omega)$ had only poles, the contour integral could simply be evaluated by residues, as in [53] and the deformed sigma models of [30, 31]. However, we have to take into account the branch cut along the imaginary axis. To do this, we move the branch cut away from the imaginary axis by a small angle $\delta$. Then, as seen in Fig. 1, the discontinuity and the poles become disentangled, and the integral along the path $\mathcal{C}$ can be separated into an integral along the discontinuity with angle $\delta$, and a sum over the residues. The resulting tilted paths, corresponding to $\delta > 0$ (respectively, $\delta < 0$) will be denoted by $\mathcal{C}_\pm$. As emphasized in [19], the value of the residues is sensitive to the sign of $\delta$, that is, to the branch choice of $\sigma(\omega)$. This leads to the renormalon ambiguity first discussed by David [12, 13]. We will denote the poles of $\sigma(i\xi)$ along the positive imaginary axis by $\xi_n$. In the $O(3)$ model they can be read from (26) and they are given by

$$\xi_n = n, \qquad n \in \mathbb{N}. \tag{29}$$

We will also denote by $\sigma_n^\pm$ the residue of $\sigma(\omega)$ at $i\xi_n \pm 0$, which reads

$$i\sigma_{2n}^\pm = \pm i\left(\frac{n}{e}\right)^{2n}\frac{2n}{(n!)^2}, \qquad i\sigma_{2n+1}^\pm = 0. \tag{30}$$

We note that the residues vanish for odd $n$, so we have removable singularities at these points.[3] One finds,

$$Q(i\xi) - \frac{1}{2\pi i}\int_{\mathcal{C}_\pm}\frac{e^{-2B\xi'}\delta\sigma(i\xi')Q(i\xi')}{\xi+\xi'}d\xi' + \sum_{n\geq 1}\frac{e^{-2B\xi_n}i\sigma_n^\pm Q_n}{\xi+\xi_n} = \frac{1}{2\pi i}\int_\mathbb{R}\frac{G_-(\omega')g_+(\omega')}{i\xi+\omega'+i0}d\omega'. \tag{31}$$

In this equation, $Q_n \equiv Q(i\xi_n)$, and

$$\delta\sigma(i\xi) = \sigma(\xi(i+0)) - \sigma(\xi(i-0)), \tag{32}$$

is the discontinuity of $\sigma(\omega)$ across the positive imaginary axis. An expression similar to (31), including exponentially small corrections, can be obtained from (17):

$$\frac{\epsilon_+(i\kappa)}{G_+(i\kappa)} = \frac{1}{2\pi i}\int_\mathbb{R}\frac{G_-(\omega')g_+(\omega')}{\omega'-i\kappa-i0}d\omega' + \frac{1}{2\pi i}\int_{\mathcal{C}_\pm}\frac{e^{-2B\kappa'}\delta\sigma(i\kappa')Q(i\kappa')}{\kappa'-\kappa}d\kappa' - \sum_{n\geq 1}\frac{e^{-2B\xi_n}i\sigma_n^\pm Q_n}{\xi_n-\kappa}. \tag{33}$$

Note that the driving term also contains exponentially small corrections. The free energy follows from (19).

In order to study the effects of the theta angle in the $O(3)$ sigma model, it is interesting to calculate explicitly the non-perturbative corrections up to second order in the exponentially small scale

$$e^{-2B} \approx e^{-4\pi/g^2(\mu)}. \tag{34}$$

This calculation is very similar to what we did in [19], and we provide some of the details in Appendix A. Here we present the results for the free energy. One finds,

$$\mathcal{F}(h) = -\frac{k^2h^2}{4}\left[\tilde{\mathcal{F}}_{(0)}(h) + \tilde{\mathcal{F}}_{(1)}(h)e^{-2B} + \tilde{\mathcal{F}}_{(2)}(h)e^{-4B} + \mathcal{O}(e^{-6B})\right] \mp i\frac{m^2}{16}, \tag{35}$$

---

[3]One could label only the even residues and neglect the odd ones ($i\sigma_{2n}^\pm = i\sigma_n'^\pm$, with $\xi_n' = 2n$). However, as we will see later, the final trans-series is organized in powers of $e^{-2nB}$ due to instanton effects. Thus, in the current numbering, $n$ tracks the exponential orders of the final trans-series.

where

$$\tilde{\mathcal{F}}_{(0)}(h) = B + \frac{\log(B) - 2 + 4\log(2)}{2} + \mathcal{O}(B^{-1}), \tag{36}$$

$$\tilde{\mathcal{F}}_{(1)}(h) = -\frac{4B^2}{e} + \frac{-2\log(B) - 3 + 2\gamma_E - 6\log(2)}{e}B + \mathcal{O}(B^0), \tag{37}$$

$$\tilde{\mathcal{F}}_{(2)}(h) = \frac{8B^2}{e^2} + \frac{4\log(B) + 2 - 4\gamma_E + 12\log(2) \mp 2i}{e^2}B + \mathcal{O}(B^0). \tag{38}$$

The imaginary ambiguity in the $e^{-4B}$ correction originates from the residue $\sigma_2^{\pm}$. The expression (35) is an example of a trans-series (see [54–56] for an overview), and it involves the small parameters $1/B$, $\log(B)/B$, and $e^{-2B}$.

As pointed out in [57], it is useful to express the free energy in terms of the coupling $\tilde{\alpha}$, defined by

$$\frac{1}{\tilde{\alpha}} + \log(\tilde{\alpha}) = \log\left(\frac{h}{\Lambda}\right). \tag{39}$$

It is easy to see that, at leading order, one has

$$\tilde{\alpha} \sim \beta_0 g^2(h), \tag{40}$$

where $g^2(h)$ is the running coupling constant at the scale $h$. One uses the boundary condition (A.21) and the mass gap (27) to obtain $B$ as a trans-series in $\tilde{\alpha}$. Then one can write (35) as

$$\mathcal{F}(h) = -\frac{h^2}{4\pi}\left[\mathcal{F}_{(0)}(h) + \mathcal{F}_{(1)}(h)e^{-2/\tilde{\alpha}} + \mathcal{F}_{(2)}(h)e^{-4/\tilde{\alpha}} + \mathcal{O}(e^{-6/\tilde{\alpha}})\right] \mp i\frac{m^2}{16}, \tag{41}$$

where

$$\begin{aligned} \mathcal{F}_{(0)}(h) &= \frac{1}{\tilde{\alpha}} - \frac{1}{2} + \mathcal{O}(\tilde{\alpha}), \\ \mathcal{F}_{(1)}(h) &= -\frac{64}{e^2\tilde{\alpha}^3} + \frac{32(-\log(\tilde{\alpha}) - 3 + \gamma_E + 5\log(2))}{e^2\tilde{\alpha}^2} + O(\tilde{\alpha}^{-1}), \\ \mathcal{F}_{(2)}(h) &= \frac{512(1 \mp i)}{e^4\tilde{\alpha}^3} + \mathcal{O}(\tilde{\alpha}^{-2}). \end{aligned} \tag{42}$$

Once again, the imaginary ambiguity in the $e^{-4/\tilde{\alpha}}$ correction originates from $\sigma_2^{\pm}$.

Finally, we write the result in the canonical formulation, i.e. in terms of the density of particles $\rho$ and energy density $e$ introduced in (8). As noted in [57], the appropriate quantity to study is the quotient $e/(\pi\rho^2)$. In addition, we will express the result in terms of the coupling $\alpha$, defined by

$$\frac{1}{\alpha} = \log\left(\frac{\rho}{2\beta_0\Lambda}\right). \tag{43}$$

One finds that $\tilde{\alpha}$ is given by a trans-series in $\alpha$, with $\tilde{\alpha} \sim \alpha$ at the very leading order. Our final result for the normalized energy density, up to second order in the exponential corrections, is

$$\begin{aligned} \frac{e}{\pi\rho^2} = \alpha + \frac{\alpha^2}{2} + \mathcal{O}(\alpha^3) + \frac{32}{e^2}\left(\frac{2}{\alpha} + \log(\alpha) + 3 - \gamma_E - 5\log(2) \mp i\frac{\pi}{2} + \frac{\alpha}{2} + \mathcal{O}(\alpha^2)\right)e^{-2/\alpha} \\ + \frac{512}{e^4}\left(\frac{1 \pm i}{\alpha} + \mathcal{O}(\alpha^0)\right)e^{-4/\alpha} + \mathcal{O}(e^{-6/\alpha}). \end{aligned} \tag{44}$$

The first ambiguous imaginary term arises from the isolated $m^2$ term in (35), while the term of order $\alpha$ in the correction proportional to $e^{-2/\alpha}$ was calculated in [24].[4] The expression (44), up to and including the first exponential correction, was tested in detail in [19, 24] against a numerical calculation based on the canonical formalism of (9), (10).

---

[4]As this paper was being typed, [58] appeared, which calculates this series up to order $\alpha^3$, and estimates numerically the coefficients up to order $\alpha^5$.

# 3  The $O(3)$ sigma model at $\vartheta = \pi$

As we mentioned in the introduction, the $O(3)$ sigma model admits a theta angle. The topological density is of the form

$$q(x) = \frac{1}{8\pi} \epsilon^{abc} \epsilon_{\mu\nu} S_a \partial_\mu S_b \partial_\nu S_c \,, \tag{45}$$

where $\epsilon^{abc}$, $\epsilon_{\mu\nu}$ are totally antisymmetric symbols for the internal and spacetime indices, respectively. The topological charge is then given by

$$\mathcal{Q} = \int q(x) \mathrm{d}^2 x \,, \tag{46}$$

and one can then add to the Euclidean action the term

$$\mathrm{i}\vartheta \mathcal{Q} \,. \tag{47}$$

The $\vartheta$ angle has important effects on the theory. Haldane famously conjectured [59] that, when $\vartheta = \pi$, the mass gap vanishes and the sigma model flows to a non-trivial conformal field theory, the $SU(2)_1$ Wess–Zumino–Witten model (see e.g. [60] for a review of these results). In particular, we can consider the free energy (3) in the presence of the theta angle,

$$\mathcal{F}(h, \vartheta) = F(h, \vartheta) - F(0, \vartheta) \,, \tag{48}$$

and we can ask how it changes as we vary $\vartheta$. Unfortunately, it is very difficult to answer this question analytically for arbitrary values of $\vartheta$. However, when $\vartheta = \pi$, the $O(3)$ non-linear sigma model is expected to be integrable, and its massless scattering theory has been proposed in [29], building on [28]. This makes it possible to calculate $\mathcal{F}(h, \pi)$. In this section we will present some detailed results for this quantity, with emphasis on its non-perturbative corrections.

## 3.1  The free energy

The general setting to calculate the free energy in a massless scattering theory has been developed in [31, 61, 62] (see [63] for a review). We consider again a conserved charge, coupled to a magnetic field or chemical potential $H$. The ground state of the massless theory is described by two rapidity distributions $\epsilon_1$ and $\epsilon_2$, which characterize right and left moving particles, respectively. The domain of these distributions is only bounded on one-side, since massless particle can have arbitrarily low momentum. The boundary condition is then

$$\epsilon_1(B) = \epsilon_2(-B) = 0 \,. \tag{49}$$

The Bethe Ansatz leads to a pair of integral equations for the rapidity distributions $\epsilon_{1,2}$, which have the form

$$\epsilon_1(\theta) - \int_{-\infty}^{B} \varphi_1(\theta - \theta') \epsilon_1(\theta') \mathrm{d}\theta' - \int_{-B}^{\infty} \varphi_2(\theta - \theta') \epsilon_2(\theta') \mathrm{d}\theta' = H - \frac{M e^{\theta}}{2} \,, \qquad \theta < B \,,$$

$$\epsilon_2(\theta) - \int_{-\infty}^{B} \varphi_2(\theta - \theta') \epsilon_1(\theta') \mathrm{d}\theta' - \int_{-B}^{\infty} \varphi_1(\theta - \theta') \epsilon_2(\theta') \mathrm{d}\theta' = H - \frac{M e^{-\theta}}{2} \,, \qquad \theta > -B \,. \tag{50}$$

In the cases we will consider, the two functions are related by parity $\epsilon_1(\theta) = \epsilon_2(-\theta)$. The kernels $\varphi_1$ and $\varphi_2$ are extracted respectively from the left-left (or right-right) massless $S$-matrix

and the left-right (or right-left) massless $S$-matrix corresponding to the particles charged under Q. In these equations, $M$ is a mass scale related to the mass gap $m$ of the theory at $\vartheta = 0$. With the above normalization, we have $M = m$. In the case of the massless theory for the $O(3)$ non-linear sigma model at $\vartheta = \pi$, the chemical potential $H$ is related to the chemical potential $h$ at $\vartheta = 0$ simply by

$$H = \frac{h}{r}, \tag{51}$$

where as shown e.g. in [63] the constant $r$ can be obtained from the kernels in (50), see (69) below for an explicit formula. The free energy is then given by

$$\mathcal{F}(h, \pi) = -\frac{M}{2\pi} \int_{-\infty}^{B} e^{\theta} \epsilon_1(\theta) d\theta. \tag{52}$$

As in the massive phase at $\vartheta = 0$, it is useful to consider the density of particles and energy introduced in (8). These can be obtained, as in (9), (10), from "dual" integral equations for Bethe root densities $\chi_1(\theta)$, $\chi_2(\theta)$,

$$\chi_1(\theta) - \int_{-\infty}^{B} \varphi_1(\theta - \theta') \chi_1(\theta') d\theta' - \int_{-B}^{\infty} \varphi_2(\theta - \theta') \chi_2(\theta') d\theta' = M e^{\theta},$$
$$\chi_2(\theta) - \int_{-B}^{\infty} \varphi_1(\theta - \theta') \chi_2(\theta') d\theta' - \int_{-\infty}^{B} \varphi_2(\theta - \theta') \chi_1(\theta') d\theta' = M e^{-\theta}. \tag{53}$$

We can bring these two integral equations into a single equation for $\chi_1(\theta)$, using the parity relation $\chi_1(-\theta) = \chi_2(\theta)$ and the symmetry of the kernels $\varphi_i(\theta) = \varphi_i(-\theta)$:

$$\chi_1(\theta) - \int_{-\infty}^{B} \left( \varphi_1(\theta - \theta') + \varphi_2(\theta + \theta') \right) \chi_1(\theta') d\theta' = M e^{\theta}. \tag{54}$$

Then the density of particles and the energy density are given by

$$e = \frac{M}{4\pi} \int_{-\infty}^{B} e^{\theta} \chi_1(\theta) d\theta, \qquad \rho = \frac{1}{2\pi r} \int_{-\infty}^{B} \chi_1(\theta) d\theta. \tag{55}$$

## 3.2 Non-perturbative corrections

We can now try to extract both the perturbative part and the non-perturbative corrections from $\mathcal{F}(h, \pi)$ in the $O(3)$ sigma model with $\vartheta = \pi$ and compare them to what we found for $\vartheta = 0$. We expect that the perturbative part remains unchanged, and we would like to understand what happens to the non-perturbative effects. These issues were already addressed in e.g. [30, 31]. However, in their calculation of the free energy, [30, 31] considered *deformations* of the conventional sigma models (including the so-called "sausage" model deformation of the $O(3)$ sigma model). These deformations have an interest of their own, and they are relatively easier to study: after the deformation, the calculation of perturbative and non-perturbative corrections becomes straightforward, as in the Sine-Gordon model studied in [53]. This is due to the fact that the analogue of the function (16) has only pole singularities, and no branch cuts. However, the deformed theories are not necessarily good guides for the undeformed theories. For example, in order to obtain the corresponding results for $\mathcal{F}(h, \pi)$ in the undeformed theory, one should perform a highly non-trivial resummation. In addition, due to the absence of branch cuts, there is no proper perturbative expansion and, in particular, no large order behavior nor renormalons. Hence an important aspect of the physics of the original models is lost.

In spite of these differences, some aspects of the analysis of the deformed theories also apply to the undeformed theories considered in this paper. As in [30, 31, 62, 63], the first step is a Wiener–Hopf analysis of the integral equations (50), similar to what is done when $\vartheta = 0$. Let us recall that one of the basic ingredients in this analysis is the additive decomposition into a function analytic in the complex upper half plane $\mathbb{H}_+$ (which carries a $+$ subscript) and a function analytic in the complex lower half plane $\mathbb{H}_-$ (which carries a $-$ subscript). For a generic function $\psi(\omega)$ defined in the real line, this can be obtained from

$$[\psi(\omega)]_\pm = \pm \frac{1}{2\pi i} \int_{\mathbb{R}} \frac{\psi(\omega')}{\omega' - (\omega \pm i0)} d\omega', \quad \omega \in \mathbb{H}_\pm. \tag{56}$$

After analytically extending each factor to the entire plane (up to singularities), we have

$$\psi(\omega) = [\psi(\omega)]_+ + [\psi(\omega)]_-. \tag{57}$$

A useful consequence of (56) is that if a function $\Psi_\pm(\omega)$ is analytic and bounded in the upper (lower) half plane, then

$$[\Psi_\pm]_\mp = 0, \qquad [e^{\pm ia\omega}\Psi_\pm]_\mp = 0, \quad a > 0. \tag{58}$$

The goal of this section is to rewrite the integral equations (50) to make the comparison with the $\vartheta = 0$ case manifest. Thus we will first reduce them to individual integral equations of the form of (31) and (33) and then single out the perturbative contributions, the non-perturbative ambiguous formally imaginary contributions and the non-perturbative formally real unambiguous contributions. As we previously discussed, we expect the first to be identical in the $\vartheta = 0$ and $\vartheta = \pi$ cases. The picture is slightly more intricate for non-perturbative corrections. Roughly speaking, imaginary ambiguous terms will remain identical, which is consistent with their connection to large order behaviour of the perturbative series, while formally real non-perturbative effects will change sign according to the rule $e^{-2nB} \to (-1)^n e^{-2nB}$. This description is complicated by the fact that real and imaginary exponential terms will sometimes multiply each other when constructing the free energy from $Q(i\xi)$ and $\epsilon_+(i\kappa)$. We will give a more precise statement by the end of this section.

While we also often omit the qualification, it is important to draw attention to the fact that these terms are only *formally* real or imaginary, i.e. we can say that they are real or imaginary in the sense that they are real or imaginary terms of a formal power series. However, these divergent formal power series are only made sense of once we apply Borel summation, which combines formally real and imaginary terms into a final, real, unambiguous result.

To start the Wiener–Hopf analysis of the integral equations (50), we have to extend their domain of validity to the full real line. We first extend the driving terms as

$$g(\theta) = \begin{cases} H - \dfrac{Me^\theta}{2}, & \text{if } \theta < B, \\ H - h, & \text{if } \theta > B, \end{cases} \tag{59}$$

and introduce an unknown function $Y(\theta)$ which is supported in the positive half-line. The constant $h$ in (59) is at the moment arbitrary and changing it amounts simply to redefining the unknown function $Y$. Then we have

$$\epsilon_1(\theta) - \int_{-\infty}^{B} \varphi_1(\theta - \theta')\epsilon_1(\theta')d\theta' - \int_{-B}^{\infty} \varphi_2(\theta - \theta')\epsilon_2(\theta')d\theta' = g(\theta) + Y(\theta - B),$$
$$\epsilon_2(\theta) - \int_{-\infty}^{B} \varphi_2(\theta - \theta')\epsilon_1(\theta')d\theta' - \int_{-B}^{\infty} \varphi_1(\theta - \theta')\epsilon_2(\theta')d\theta' = g(-\theta) + Y(-\theta - B). \tag{60}$$

To find the free energy of (52), we will first reduce the two above equations into a single equation for $\epsilon_1$. To this end, we take the Fourier transform of the integral equations, which gives

$$\tilde{\epsilon}_1 - \phi_1 \tilde{\epsilon}_1 - \phi_2 \tilde{\epsilon}_2 = e^{iB\omega} Y_+ + e^{iB\omega} g_-^m + H 2\pi \delta - h \frac{i e^{iB\omega}}{\omega + i0}, \tag{61}$$

$$\tilde{\epsilon}_2 - \phi_2 \tilde{\epsilon}_1 - \phi_1 \tilde{\epsilon}_2 = e^{-iB\omega} Y_- + e^{-iB\omega} g_+^m + H 2\pi \delta + h \frac{i e^{-iB\omega}}{\omega - i0}, \tag{62}$$

where $\tilde{\epsilon}_i$, $\phi_i$ and $Y_+$ are the Fourier transforms of $\epsilon_i$, $\varphi_i$ and $Y$, respectively. The last three terms in each equation correspond to the Fourier transform of $g(\theta)$ and $g(-\theta)$, respectively, where we have conveniently isolated the Fourier transform of the term proportional to $M$ in (59):

$$g_-^m(\omega) = e^{-iB\omega} \int_{-\infty}^{B} e^{i\omega\theta} \left( -\frac{m e^\theta}{2} \right) d\theta = \frac{m e^B}{2} \frac{i}{\omega - i}. \tag{63}$$

From now on, we will use that $M = m$. Finally, $Y_-(\omega) = Y_+(-\omega)$, $g_+^m(\omega) = g_-^m(-\omega)$ and $\delta$ is the Dirac delta function. We can now solve for $\tilde{\epsilon}_2$ in (62), which gives

$$\tilde{\epsilon}_2 = \frac{1}{1 - \phi_1} \left( \phi_2 \tilde{\epsilon}_1 + e^{-iB\omega} Y_- + e^{-iB\omega} g_+^m + H 2\pi \delta + h \frac{i e^{-iB\omega}}{\omega - i0} \right). \tag{64}$$

If we plug this expression back into (61), we obtain

$$\left( 1 - \phi_1 - \frac{\phi_2^2}{1 - \phi_1} \right) \epsilon_- = Y_+ + g_-^m + e^{-2iB\omega} \frac{\phi_2}{1 - \phi_1} \left( Y_- + g_+^m \right)$$
$$+ H 2\pi \delta \left( 1 + \frac{\phi_2(0)}{1 - \phi_1(0)} \right) - h \left( \frac{i}{\omega + i0} - \frac{\phi_2}{1 - \phi_1} \frac{i e^{-2iB\omega}}{\omega - i0} \right), \tag{65}$$

where

$$\epsilon_-(\omega) = e^{-iB\omega} \tilde{\epsilon}_1(\omega). \tag{66}$$

The l.h.s. of (65) suggests to consider the Wiener–Hopf factorization of the kernel

$$1 - \phi_1(\omega) - \frac{\phi_2^2(\omega)}{1 - \phi_1(\omega)} = \frac{1}{K_+(\omega) K_-(\omega)}, \tag{67}$$

where $K_\pm(\omega)$ is analytic in the upper (respectively, lower) complex half plane. We now use the additive decomposition of the $\delta$-function

$$2\pi \delta(\omega) = \frac{i}{\omega + i0} - \frac{i}{\omega - i0}, \tag{68}$$

and we see that the choice

$$h = H \left( 1 + \frac{\phi_2(0)}{1 - \phi_1(0)} \right), \tag{69}$$

simplifies the terms singular at the origin. This $h$ will be eventually identified with the chemical potential of the theory with $\vartheta = 0$, therefore (69) provides the relation (51) stated above. Lastly, we define

$$v = K_+ Y_+, \tag{70}$$

which plays the role of the function $Q$ appearing in (15). Algebraic manipulations then give us

$$\frac{\epsilon_-}{K_-} = v(\omega) + e^{-2iB\omega} \frac{\phi_2}{1 - \phi_1} \sigma(-\omega) v(-\omega) + K_+ \left( g_-^m + e^{-2iB\omega} \frac{\phi_2}{1 - \phi_1} g_+^m \right)$$
$$- \frac{i h K_+}{\omega - i0} \left( 1 - e^{-2iB\omega} \frac{\phi_2}{1 - \phi_1} \right), \tag{71}$$

where

$$\sigma(\omega) = \frac{K_-(\omega)}{K_+(\omega)}. \tag{72}$$

So far our considerations have been very general. Let us now focus on the specific example of the $O(3)$ sigma model with $\vartheta = \pi$. After turning on the chemical potential as in (22), one finds a pair of equations of the form (50), where the Fourier transforms of the kernels are given by [31]

$$\phi_1(\omega) = \phi_2(\omega) = \frac{1}{1 + e^{\pi|\omega|}}. \tag{73}$$

It turns out that

$$1 - \phi_1(\omega) - \frac{\phi_2^2(\omega)}{1 - \phi_1(\omega)} = 1 - \tilde{K}(\omega), \tag{74}$$

where $\tilde{K}(\omega)$ is defined in (25). Therefore, $K_\pm(\omega) = G_\pm(\omega)$, where $G_+(\omega)$ is given in (26). Furthermore, we find the relation

$$\frac{\phi_2}{1 - \phi_1} = 1 - \frac{1}{G_+ G_-}. \tag{75}$$

We now take the $(+)$-projection of (71). Due to the simplification (75) we can cancel many terms using the property (58), and we find

$$v = -[G_+ g_-]_+ - \left[e^{-2iB\omega}\{\sigma v\}(-\omega)\right]_+ - \frac{ime^{-B}}{2G_+(i)}\frac{1}{\omega + i}, \tag{76}$$

where we have introduced

$$g_+(\omega) = g_-(-\omega) = g_+^{(p)}(\omega) + \frac{ime^B}{2}\frac{e^{2iB\omega}}{\omega - i}, \tag{77}$$

and

$$g_+^{(p)}(\omega) = ih\frac{1 - e^{2iB\omega}}{\omega} - \frac{ime^B}{2}\frac{1}{\omega + i}. \tag{78}$$

Note that, although we keep a subscript $+$ for both, $g_+^{(p)}$ is analytic in the upper half plane, but $g_+$ is not.

One can also obtain an equation for $\epsilon_-$ from (71). After taking the $(-)$-projection, we find

$$\frac{\epsilon_-}{G_-} = [G_+ g_-]_- + \left[e^{-2iB\omega}\{\tilde{\sigma} v\}(-\omega)\right]_- + \frac{ime^B}{2(\omega + i)}\left(\frac{e^{-2iB\omega}}{G_-(\omega)} - \frac{e^{-2B}}{G_+(i)}\right) - \frac{ihe^{-2iB\omega}}{\omega G_-(\omega)}, \tag{79}$$

where we now introduce

$$\tilde{\sigma}(\omega) = \frac{G_-(\omega)}{G_+(\omega)} - \frac{1}{G_+^2(\omega)}. \tag{80}$$

We remark that the extra term $1/G_+^2(\omega)$ is analytic in the upper half plane, which will have important consequences. In particular, it will imply that the perturbative part does not change when going from $\vartheta = 0$ to $\vartheta = \pi$.

In order to make contact with the analysis of [19], it is useful to write (76) and (79) in integral form. For $v(\omega)$ we have

$$v(i\xi) = \frac{1}{2\pi i}\int_{\mathbb{R}}\frac{G_-(\omega')g_+(\omega')}{i\xi + \omega' + i0}d\omega' + \frac{1}{2\pi i}\int_{\mathbb{R}}\frac{e^{2iB\omega'}\sigma(\omega')v(\omega')}{i\xi + \omega' + i0}d\omega' - \frac{me^{-B}}{2G_+(i)}\frac{1}{\xi + 1}, \tag{81}$$

and for $\epsilon$ we get

$$
\frac{\epsilon_+(i\kappa)}{G_+(i\kappa)} = \frac{1}{2\pi i}\int_{\mathbb{R}}\frac{G_-(\omega')g_+(\omega')}{\omega'-i\kappa-i0}d\omega' + \frac{1}{2\pi i}\int_{\mathbb{R}}\frac{e^{2iB\omega'}\tilde{\sigma}(\omega')v(\omega')}{\omega'-i\kappa-i0}d\omega'
$$
$$
- \frac{me^B}{2(\kappa-1)}\left(\frac{e^{-2B\kappa}}{G_+(i\kappa)} - \frac{e^{-2B}}{G_+(i)}\right) + \frac{he^{-2B\kappa}}{\kappa G_+(i\kappa)}. \quad (82)
$$

Equations (81) and (82) are respectively the counterparts in the massless theory of equations (15) and (17) in the massive theory. We can analyze them as in [19], and extract the non-perturbative corrections by deforming the contour as we did for the theory at $\vartheta = 0$.

Let us first consider (81). The two integrals contain perturbative terms due to the discontinuity, non-perturbative imaginary terms due to the poles of $G_-$ (and thus $\sigma$) while only the first integral has a non-perturbative real contribution from the pole of $g_+$ at $\omega = i$. We want to single out this last contribution. Since $g_+^{(p)}$ does not have such a pole, we expand the first integral in (81) and plug in

$$
G_-(i) = \frac{1}{\sqrt{2e}}, \qquad G_+(i) = \sqrt{\frac{e}{2}}. \quad (83)
$$

We then find

$$
v(i\xi) = \frac{1}{2\pi i}\int_{\mathbb{R}}\frac{G_-(\omega')g_+^{(p)}(\omega')}{i\xi+\omega'+i0}d\omega' + \frac{1}{2\pi i}\int_{\mathbb{R}}\frac{e^{2iB\omega'}\sigma(\omega')v(\omega')}{i\xi+\omega'+i0}d\omega'
$$
$$
+ \frac{me^B}{4\pi i}\int_{\mathcal{C}_\pm}\frac{e^{-2B\xi'}\delta G_-(i\xi')}{(\xi+\xi')(\xi'-1)}d\xi' - \frac{me^B}{2}\sum_{n\geq 1}\frac{e^{-2B\xi_n}G_+(i\xi_n)i\sigma_n^\pm}{(\xi_n+\xi)(\xi_n-1)} - \frac{me^{-B}}{2\sqrt{2e}}\frac{1}{\xi+1}. \quad (84)
$$

In this form it is easier to break down the several contributions. Since $g_+^{(p)}$ and $v$ are analytic in the upper half plane, we can deform the first two integrals around the positive imaginary axis, which returns a perturbative part due to the discontinuity, and non-perturbative imaginary terms due to the poles. The third term in (84) is already an integral along the discontinuity which can be expanded into a perturbative series. The terms $G_+(i\xi_n)\sigma_n^\pm$ in the sum are imaginary and ambiguous, while the last term in (84) is the sole real unambiguous non-perturbative term.

For $\vartheta = 0$, a similar expansion of (31) gives the following analogous formula:

$$
Q(i\xi) = \frac{1}{2\pi i}\int_{\mathbb{R}}\frac{G_-(\omega')g_+^{(p)}(\omega')}{i\xi+\omega'+i0}d\omega' + \frac{1}{2\pi i}\int_{\mathbb{R}}\frac{e^{2iB\omega'}\sigma(\omega')Q(\omega')}{i\xi+\omega'+i0}d\omega'
$$
$$
+ \frac{me^B}{4\pi i}\int_{\mathcal{C}_\pm}\frac{e^{-2B\xi'}\delta G_-(i\xi')}{(\xi+\xi')(\xi'-1)}d\xi' - \frac{me^B}{2}\sum_{n\geq 1}\frac{e^{-2B\xi_n}G_+(i\xi_n)i\sigma_n^\pm}{(\xi_n+\xi)(\xi_n-1)} + \frac{me^{-B}}{2\sqrt{2e}}\frac{1}{\xi+1}. \quad (85)
$$

Comparing (84) and (85) with $Q \leftrightarrow v$, we see that only the real exponentially suppressed contribution has the opposite sign. While it is not apparent from this derivation, it is useful to keep in mind that

$$
h \sim me^B, \quad (86)
$$

and thus $me^{-B}$ is a correction proportional to $e^{-2B}$ once it has been expressed in terms of $h$.

Let us do a similar analysis of the equation for $\epsilon_+(i\kappa)$, (82). This equation is used in the calculation of the free energy in two distinct ways. First, to compute the free energy proper in (52) from the Wiener–Hopf solution. In this case, we evaluate (82) at $\kappa = 1$ to obtain

$$
\mathcal{F}(h,\pi) = -\frac{1}{2\pi}me^B\epsilon_+(i). \quad (87)
$$

Second, to compute the relation between $B$ and $m/h$. In this case, we impose the boundary condition (49), which in Fourier space takes the form:

$$\lim_{\kappa \to \infty} \kappa \epsilon_+(i\kappa) = 0.\tag{88}$$

Thus, we have to compute the limit of (82) as $\kappa \to \infty$. These two evaluations of $\epsilon_+(i\kappa)$, at $\kappa = 1$ and in the limit $\kappa \to \infty$, must be analyzed separately.

Let us start with the $\kappa = 1$ case by observing that the second term in the r.h.s. of (82) can be written as

$$\frac{1}{2\pi i}\int_{\mathbb{R}}\frac{e^{2iB\omega'}\tilde{\sigma}(\omega')v(\omega')}{\omega'-i-i0}d\omega' = \frac{1}{2\pi i}\int_{\mathcal{C}_\pm}\frac{e^{-2B\xi'}\delta\sigma(i\xi')v(i\xi')}{\xi'-1}d\xi' - \sum_{n\geq 1}\frac{e^{-2B\xi_n}i\sigma_n^\pm v(i\xi_n)}{\xi_n-1}$$
$$-\frac{e^{-2B}}{e}v(i),\tag{89}$$

where we use that discontinuities and residues of $\tilde{\sigma}$ are identical to those of $\sigma$, since they differ only by the addition of the function $1/G_+^2(\omega)$, which is analytic in the upper half plane, as we remarked in (80). This formula contrasts with the analogous integral in the $\vartheta = 0$ calculation:

$$\frac{1}{2\pi i}\int_{\mathbb{R}}\frac{e^{2iB\omega'}\sigma(\omega')Q(\omega')}{\omega'-i-i0}d\omega' = \frac{1}{2\pi i}\int_{\mathcal{C}_\pm}\frac{e^{-2B\xi'}\delta\sigma(i\xi')Q(i\xi')}{\xi'-1}d\xi' - \sum_{n\geq 1}\frac{e^{-2B\xi_n}i\sigma_n^\pm Q(i\xi_n)}{\xi_n-1}$$
$$+\frac{e^{-2B}}{e}Q(i).\tag{90}$$

Thus for this term we have once again that the perturbative and imaginary non-perturbative terms are identical, but the real non-perturbative term changes sign.

Furthermore, in the first integral of (82), we also have real exponentially suppressed terms, which come from the residue of the double pole at $\omega = i$, and they are identical to the $\vartheta = 0$ case. Explicitly, we find

$$\frac{1}{2\pi i}\int_{\mathbb{R}}\frac{G_-(\omega')g_+(\omega')}{\omega'-i}d\omega' = (\text{perturbative terms}) - \sum_{n\geq 1}\frac{e^{-2B\xi_n}G_+(i\xi_n)i\sigma_n^\pm g_n}{\xi_n-1}$$
$$-he^{-2B}G_-(i) - \frac{me^{-B}}{2}\left(2BG_-(i)+iG_-'(i\pm 0)\right),\tag{91}$$

where

$$g_n = -\frac{h}{\xi_n} + \frac{me^B}{2(\xi_n-1)}.\tag{92}$$

For the particular value of $\kappa = 1$, the non-integrated terms in (82) simplify to

$$\lim_{\kappa \to 1}\left[\frac{he^{-2B\kappa}}{\kappa G_+(i\kappa)} - \frac{me^B}{2(\kappa-1)}\left(\frac{e^{-2B\kappa}}{G_+(i\kappa)} - \frac{e^{-2B}}{G_+(i)}\right)\right] = h\frac{e^{-2B}}{G_+(i)} + \frac{me^{-B}}{2}\left(\frac{2B}{G_+(i)} + \frac{iG_+'(i)}{G_+^2(i)}\right).\tag{93}$$

Using the values $G_+(i)$ and $G_-(i)$ in (83), we can see that the above contributions change the sign in the real part of the two terms appearing in last line of (91), while keeping their ambiguous imaginary part unmodified. More explicitly, when putting everything together, we find for the $\vartheta = \pi$ case

$$\frac{\epsilon_+(i)}{G_+(i)} = (\text{perturbative terms}) - \sum_{n\geq 1}\frac{e^{-2B\xi_n}i\sigma_n^\pm v(i\xi_n)}{\xi_n-1} - \sum_{n\geq 1}\frac{e^{-2B\xi_n}G_+(i\xi_n)i\sigma_n^\pm g_n}{\xi_n-1}$$
$$-\frac{e^{-2B}}{e}v(i) + he^{-2B}\sqrt{\frac{1}{2e}} + \frac{me^{-B}}{2}\left(\sqrt{\frac{2}{e}}B - \frac{1}{2\sqrt{2e}}(-1+\gamma_E+\log 2)\right) \pm \frac{i\pi me^{-B}}{4\sqrt{2e}}.\tag{94}$$

This contrasts with the $\vartheta = 0$ case

$$\frac{\epsilon_+(\mathrm{i})}{G_+(\mathrm{i})} = \text{(perturbative terms)} - \sum_{n\geq 1}\frac{\mathrm{e}^{-2B\xi_n}\mathrm{i}\sigma_n^\pm Q(\mathrm{i}\xi_n)}{\xi_n-1} - \sum_{n\geq 1}\frac{\mathrm{e}^{-2B\xi_n}G_+(\mathrm{i}\xi_n)\mathrm{i}\sigma_n^\pm g_n}{\xi_n-1}$$
$$+ \frac{\mathrm{e}^{-2B}}{\mathrm{e}}Q(\mathrm{i}) - h\mathrm{e}^{-2B}\sqrt{\frac{1}{2\mathrm{e}}} - \frac{m\mathrm{e}^{-B}}{2}\left(\sqrt{\frac{2}{\mathrm{e}}}B - \frac{1}{2\sqrt{2\mathrm{e}}}(-1+\gamma_E+\log 2)\right) \pm \frac{\mathrm{i}\pi m\mathrm{e}^{-B}}{4\sqrt{2\mathrm{e}}}, \quad (95)$$

where the perturbative terms are the same in both cases up to $Q \leftrightarrow v$.

Finally, in the $\kappa \to \infty$ limit, which we need to impose the boundary condition, the non-integrated terms with exponential dependency on $\kappa$ do not contribute. From $\lim_{\kappa\to\infty} G_+(\mathrm{i}\kappa) = 1$, we find that the limit in the $\vartheta = \pi$ case is then given by

$$\lim_{\kappa\to\infty} \kappa\epsilon_+(\mathrm{i}\kappa) = \text{(perturbative terms)} + \sum_{n\geq 1}\mathrm{e}^{-2B\xi_n}\mathrm{i}\sigma_n^\pm(v(\mathrm{i}\xi_n) + G_+(\mathrm{i}\xi_n)g_n) + \frac{m\mathrm{e}^{-B}}{2\sqrt{2\mathrm{e}}}, \quad (96)$$

which contrasts with the $\vartheta = 0$ case

$$\lim_{\kappa\to\infty} \kappa\epsilon_+(\mathrm{i}\kappa) = \text{(perturbative terms)} + \sum_{n\geq 1}\mathrm{e}^{-2B\xi_n}\mathrm{i}\sigma_n^\pm(Q(\mathrm{i}\xi_n) + G_+(\mathrm{i}\xi_n)g_n) - \frac{m\mathrm{e}^{-B}}{2\sqrt{2\mathrm{e}}}. \quad (97)$$

By comparing equations (84), (94) and (96) with equations (85), (95) and (97), we see that one can transform the integral equations for the problem with $\vartheta = \pi$ into the $\vartheta = 0$ equations by swapping $Q \leftrightarrow v$ and $\mathrm{e}^{-2B} \to -\mathrm{e}^{-2B}$ in the non-ambiguous terms, i.e. those not multiplied by $\mathrm{i}\sigma_n^\pm$. This is true at the level of the equations and at leading order, but we can see that the solution does not map as straightforwardly. For example, when solving for $v(\mathrm{i}\xi)$ there is a term $\mathrm{e}^{-2B\xi_n}\mathrm{i}\sigma_n^\pm v(\mathrm{i}\xi_n)$ which does not change sign under this map, but $v(\mathrm{i}\xi_m)$ itself has a real non-perturbative correction of order $\mathrm{e}^{-2B}$ which does change sign. Furthermore, once we impose the boundary condition and re-express $B$ in terms of $h$, or alternatively $\tilde{\alpha}(h)$, the terms which swap and do not swap sign become further entangled. However, one can easily keep track of them by, for example, multiplying the appropriate terms by a factor $\cos(\vartheta)$ in the original equations.

From this prescription, we get the following result for the normalized energy density at $\vartheta = \pi$:

$$\frac{e}{\pi\rho^2} = \alpha + \frac{\alpha^2}{2} + \mathcal{O}(\alpha^3) - \frac{32}{\mathrm{e}^2}\left(\frac{2}{\alpha} + \log(\alpha) + 3 - \gamma_E - 5\log(2) \pm \frac{\mathrm{i}\pi}{2} + \frac{\alpha}{2} + \mathcal{O}(\alpha^2)\right)\mathrm{e}^{-2/\alpha}$$
$$+ \frac{512}{\mathrm{e}^4}\left(\frac{1\pm\mathrm{i}}{\alpha} + \mathcal{O}(\alpha^0)\right)\mathrm{e}^{-4/\alpha} + \mathcal{O}(\mathrm{e}^{-6/\alpha}), \quad (98)$$

which is the expression we previously obtained for the $\vartheta = 0$ case, in (44), but changing the sign of the real part in the $\mathrm{e}^{-2/\alpha}$ correction. Let us recall that, in the presence of a theta angle, an instanton configuration with topological charge $\mathcal{Q}$ in the $O(3)$ non-linear sigma model contributes to a physical observable as

$$\exp\left\{\mathrm{i}\mathcal{Q}\vartheta - \frac{4\pi|\mathcal{Q}|}{g^2(\mu)}\right\}. \quad (99)$$

In particular, an instanton configuration with an odd value of $\mathcal{Q}$ will change sign as we go from $\vartheta = 0$ to $\vartheta = \pi$. The change of sign of the real, exponentially small correction of order $\mathrm{e}^{-2/\alpha}$ in (98) is precisely what is expected from a configuration with $\mathcal{Q} = \pm 1$ (i.e. an (anti)-one-instanton). The non-ambiguous part of the correction of order $\mathrm{e}^{-4/\alpha}$, which does not change

sign, would correspond to an instanton correction with $\mathcal{Q} = \pm 2$. More generally, the change of sign $\mathrm{e}^{-2B} \to -\mathrm{e}^{-2B}$ in the real contributions is the expected behavior of instanton corrections.

As was discussed in [19], one of the important roles of the formally imaginary ambiguous terms $\mathrm{i}\sigma_n^{\pm}$ is that they cancel the ambiguities originating in the resummation of the divergent perturbative part. It is thus important that they do not change sign between the two cases, since the perturbative part itself remains identical. On the other hand, this also means that the terms that do swap sign are *a priori* invisible to the perturbative series.

We want to remark that even if the residues $\mathrm{i}\sigma_n^{\pm}$ do not change sign, it does not follow that formally imaginary ambiguous terms do not change sign in the final result. As we mentioned earlier, terms like $\mathrm{e}^{-2B\xi_n}\mathrm{i}\sigma_n^{\pm}v(\mathrm{i}\xi_n)$ will give rise to imaginary ambiguous terms that do change sign. This is still consistent with the cancellation of imaginary ambiguities in the Borel resummation of the whole trans-series. Indeed, if an exponential sector has a real part that changes sign, then the imaginary ambiguity arising from the Borel sum of that exponential sector has to change sign accordingly.

Let us note that our result for the change in the free energy as we vary the theta angle from $\vartheta = 0$ to $\vartheta = \pi$ suggests the following instanton-type dependence of the free energy on the theta angle $\vartheta$,

$$\mathcal{F}(h, \vartheta) = -\frac{h^2}{4\pi}\mathcal{F}_{(0)}(h) \mp \mathrm{i}\frac{m^2}{16} - \cos(\vartheta)\frac{h^2}{4\pi}\mathcal{F}_{(1)}(h)\mathrm{e}^{-2/\tilde{\alpha}} + \mathcal{O}\left(\mathrm{e}^{-4/\tilde{\alpha}}\right), \tag{100}$$

where $\mathcal{F}_{(0)}(h)$ and $\mathcal{F}_{(1)}(h)$ are given in (42), and the term $\cos(\vartheta)$ comes from adding the contributions of the instantons with $\mathcal{Q} = \pm 1$. It is then natural to define an $h$-dependent topological susceptibility

$$\chi_t(h) = \left(\frac{\mathrm{d}^2\mathcal{F}(h, \vartheta)}{\mathrm{d}\vartheta^2}\right)_{\vartheta = 0}. \tag{101}$$

Therefore, we have the following asymptotic result as $\tilde{\alpha} \to 0$:

$$\chi_t(h) \approx \frac{h^2}{4\pi}\left(\frac{64}{\mathrm{e}^2\tilde{\alpha}^3} - \frac{32(-\log(\tilde{\alpha}) - 3 + \gamma_E + 5\log(2))}{\mathrm{e}^2\tilde{\alpha}^2} + O\left(\tilde{\alpha}^{-1}\right)\right)\mathrm{e}^{-2/\tilde{\alpha}}. \tag{102}$$

This is a genuine non-perturbative result for the $O(3)$ sigma model which may be testable in lattice computations (lattice calculations for the $O(3)$ sigma model with an $h$ field have been considered in e.g. [64]).

## 3.3 Testing the analytic results

In the following section, we will give ample numerical evidence that the computations of sections 2.2 and 3.2 led to the correct non-perturbative corrections for the $O(3)$ sigma model at $\vartheta = 0$ and $\pi$. To this end, we compute numerically the normalized energy density $e/(\pi\rho^2)$ and the associated $\alpha$ directly from the Bethe Ansatz equation in their canonical formulation. The coupling $\alpha$ can be obtained from its definition in (43), together with the mass gap (27), once $\rho$ has been computed.

For $\vartheta = 0$, we solve numerically the integral equation (9) for $\chi(\theta)$. This can be done straightforwardly by discretizing the integral equation with a Gauss–Kronrod quadrature in the integration interval $[-B, B]$, and solving the resulting system of equations for $\chi(\theta)$. Then we compute $\left(\alpha, e/(\pi\rho^2)\right)$ through the integrals of (55), using the same quadrature rule. One can achieve very high precision in the computation by using discretizations of the integral equation with a relatively small number of points.

For $\vartheta = \pi$ the situation is very different. The integral equation (54) has an unbounded integration interval, and the kernel decays slowly, like $1/x^2$. This leads to a substantial loss of precision. In order to achieve the accuracy required to detect exponentially small corrections,

one needs a detailed analysis of the integral equation, and the development of appropriate numerical tools, which we explain in Appendix B.

Once we have obtained $\left(\alpha, e/(\pi\rho^2)\right)$ from the Bethe Ansatz equations, we need to subtract the perturbative part, so we can test our prediction for the first exponential correction. This requires some tools from the theory of resurgent asymptotics which we now summarize briefly, see [54–56, 65] for details and references. Let us write the perturbative part of the normalized energy density as

$$\frac{e}{\pi\rho^2} \sim \alpha\varphi(\alpha), \qquad \varphi(\alpha) = \sum_{k\geq 0} e_k \alpha^k. \tag{103}$$

We wrote the first two terms of this formal power series in (44) and (98). In order to resum this series, we introduce its Borel transform

$$\widehat{\varphi}(\zeta) = \sum_{k\geq 0} \frac{e_k}{k!}\zeta^k, \tag{104}$$

which has singularities on the positive real axis, as shown in [7, 9, 19]. We also introduce the (lateral) Borel resummations of $\varphi(\alpha)$ as

$$s_\pm(\varphi)(\alpha) = \frac{1}{\alpha}\int_{\mathcal{C}_\pm} \widehat{\varphi}(\zeta)e^{-\zeta/\alpha}d\zeta, \tag{105}$$

where $\mathcal{C}_\pm$ is a straight line from 0 to $\infty$ slightly above or below the positive real line. The Borel sum will have imaginary ambiguities due to the singularities in the Borel transform, and these cancel with the explicit imaginary ambiguities appearing in the exponential corrections of (44) and (98). This cancellation was tested explicitly in [19] for the ambiguous term in the $e^{-2/\alpha}$ correction and we checked with great precision that this also happens for the $e^{-4/\alpha}$ correction.

In Fig. 2, we show the points $\left(\alpha, e/(\pi\rho^2)\right)$ that we obtained by solving numerically the Bethe Ansatz integral equations for the $O(3)$ sigma model, for both $\vartheta = 0$ and $\pi$. When $\vartheta = \pi$, we used the values $B = 10/k$, with integers $1 \leq k \leq 10$. When $\vartheta = 0$, we looked for the values of $B$ that would result into the same numerical $\alpha$ that we found for $\vartheta = \pi$.[5] This can be done with a simple bisection method up to desired precision. The dashed line corresponds to $\alpha\,\mathrm{Re}\left[s_\pm(\varphi)(\alpha)\right]$. To obtain the Borel resummation of the perturbative series, we used 70 coefficients, computed with the generalization of Volin's method [7–9], and we improved the numerical result with a conformal mapping, a strategy similar to the "Padé–Conformal–Borel" method of [66]. The error estimates are extracted from the convergence of the Padé approximant, see [6, 19]. One can see that $e/(\pi\rho^2)$ coalesces to the same perturbative result for both values of the $\vartheta$ angle, when $\alpha$ is small (corresponding to large $B$). However, as $\alpha$ grows (i.e. $B$ gets smaller), the two results start to differ, due to the different exponential corrections to the perturbative expansion.

We can test the $e^{-2/\alpha}$ correction in our results by considering the difference between $e/(\pi\rho^2)$ and the real part of the Borel sum. According to (44) and (98), this difference should have the asymptotic expansion

$$\frac{e}{\pi\rho^2} - \alpha\,\mathrm{Re}\left[s_\pm(\varphi)(\alpha)\right] \sim \pm\frac{32}{e^2}\left(\frac{2}{\alpha} + \log(\alpha) + 3 - \gamma_E - 5\log(2) + \frac{\alpha}{2} + \mathcal{O}(\alpha^2)\right)e^{-2/\alpha} + \mathcal{O}\left(e^{-4/\alpha}\right), \tag{106}$$

where the $+$ and $-$ sign correspond to $\vartheta = 0$ and $\pi$, respectively. We have also included the term of order $\alpha$ that was calculated in [24]. The asymptotic expansion for $\vartheta = 0$ was already tested in [19, 24], so we focus our attention on the $\vartheta = \pi$ case.

---

[5]Because the relation between $\alpha$ and $B$ includes exponential corrections, it is distinct between the two values of $\vartheta$.

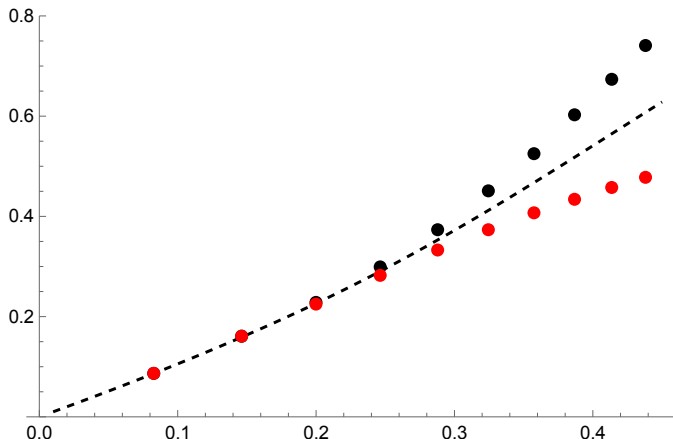

Figure 2: Numerical evaluation of $\left(\alpha, e/(\pi\rho^2)\right)$ for the $O(3)$ sigma model with $\vartheta = 0$ (black dots) and $\vartheta = \pi$ (red dots). The dashed line corresponds to the Borel resummation $\alpha\,\mathrm{Re}\left[s_{\pm}(\varphi)(\alpha)\right]$ of the perturbative part, which is shared by both values of the $\vartheta$ angle.

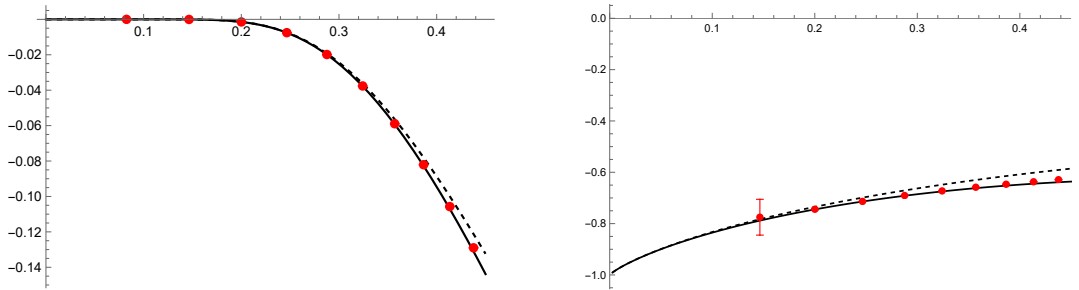

Figure 3: In the figure on the left we plot the difference between the normalized energy density $e/(\pi\rho^2)$ and the real part of the Borel resummation of the perturbative series, for the $O(3)$ sigma model at $\vartheta = \pi$. The dots are the numerical calculations of the difference. The two lines are the theoretical prediction, given by the asymptotic expansion in the r.h.s. of (106), including terms up to order $\alpha^0$ (dashed) and $\alpha$ (solid). In the figure on the right, the result has been normalized by dividing by $\mathrm{e}^{-2/\alpha}\,64/(\mathrm{e}^2\alpha)$.

In Fig. 3, we compare the numerical value of $e/(\pi\rho^2) - \alpha\,\mathrm{Re}\left[s_{\pm}(\varphi)(\alpha)\right]$ against the asymptotic expansion of (106), for $\vartheta = \pi$. The same information is shown in Table 1. In general, we find a very good agreement. For large values of $\alpha$ there is a slight disagreement that can be accounted for missing terms in the asymptotic expansion. For the points with small $\alpha$, there is also a slight disagreement (in absolute terms) that can only be appreciated in the plot on the right, in Fig. 3.[6] In this case, the error originates in the numerical computation of $e$ and $\rho$, and it could be reduced by allocating more computer time into the integral equations of the Bethe Ansatz. The error bars have been estimated from the convergence of $\rho$ when increasing the order of the integral equation quadrature, as explained at the end of Appendix B.

We also tested the $\mathrm{e}^{-4/\alpha}$ correction in (44) and (98). As we do not have enough terms in the $\alpha$ expansion accompanying $\mathrm{e}^{-2/\alpha}$, it is not possible to subtract the whole exponential correction and still be left with enough precision to test the subleading exponential correction.

---

[6]The point with lowest $\alpha$ is completely off in this plot, and it does not show inside the displayed range, although its estimated error is similarly very large. Small numerical errors become large when multiplying by $\mathrm{e}^{2/\alpha}$, with $\alpha$ small, thus very high numerical precision is required.

Table 1: Difference between the normalized energy density $e/(\pi\rho^2)$ and the real part of the Borel resummation of the perturbative series, for the $O(3)$ sigma model at $\vartheta = \pi$. We compare the numerical value obtained from the Bethe Ansatz (2nd column), to the asymptotic expansion in the r.h.s. of (106) (3rd column).

| $\alpha$ | Numerical result | Asymptotics |
|---|---|---|
| 0.082486 | $(0.5\pm2.6) \times 10^{-6}$ | $-2.65089 \times 10^{-9}$ |
| 0.146461 | $-0.00005(4\pm5)$ | $-0.000055$ |
| 0.200073 | $-0.00146(8\pm7)$ | $-0.001469$ |
| 0.246564 | $-0.00752(2\pm9)$ | $-0.007526$ |
| 0.287605 | $-0.0198(57\pm10)$ | $-0.019884$ |
| 0.324195 | $-0.0376(11\pm11)$ | $-0.037725$ |
| 0.357020 | $-0.0589(40\pm12)$ | $-0.059259$ |
| 0.386608 | $-0.0820(84\pm13)$ | $-0.082778$ |
| 0.413390 | $-0.1057(27\pm14)$ | $-0.106994$ |
| 0.437725 | $-0.1290(03\pm15)$ | $-0.131052$ |

However, one way to circumvent this issue is to consider instead the sum of $e/(\pi\rho^2)$ at $\vartheta = 0$ and $\vartheta = \pi$. According to (44) and (98), the correction of order $e^{-2/\alpha}$ cancels in the sum, leading to

$$\frac{1}{2\pi}\left[\frac{e}{\rho^2}(\alpha,0) + \frac{e}{\rho^2}(\alpha,\pi)\right] - \alpha \operatorname{Re}\left[s_\pm(\varphi)(\alpha)\right] \sim \left(\frac{512}{e^4\alpha} + \mathcal{O}(\alpha^0)\right)e^{-4/\alpha} + \mathcal{O}(e^{-8/\alpha}). \quad (107)$$

We test this expression in Fig. 4, finding again good agreement.

## 4 Fendley's integrable models

### 4.1 General aspects

Due to the integrability of the $O(3)$ sigma model at $\vartheta = 0$ and $\vartheta = \pi$, we have been able to study the effects of the theta angle on the non-perturbative effects appearing in the free energy. It

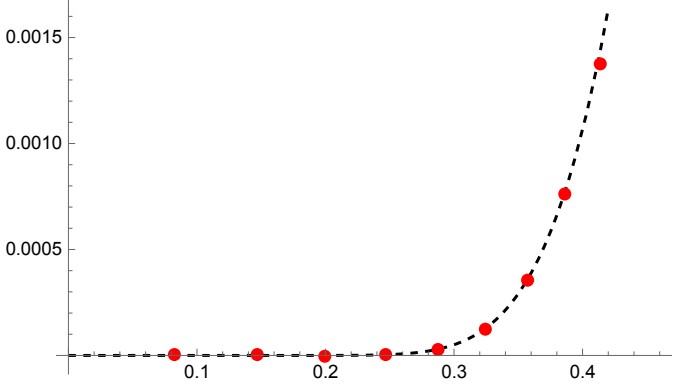

Figure 4: Difference between $\left[e/\rho^2(\alpha,0) + e/\rho^2(\alpha,\pi)\right]/(2\pi)$ and the real part of the Borel resummation of the perturbative series, for the $O(3)$ sigma model. The dots are the numerical calculations of the difference. The dashed line is the theoretical prediction, given by the asymptotic expansion in the r.h.s. of (107).

is natural to ask whether this feature can be generalized to other models. This was answered in the affirmative in [30]. The coset sigma models $SU(N)/SO(N)$ and $O(2P)/O(P) \times O(P)$ have $\pi_1(G/H) = \mathbb{Z}_2$ (for $N, P > 2$), so there is a $\mathbb{Z}_2$ topological charge and one can introduce a discrete theta angle which can only take the values $\vartheta = 0, \pi$. Fendley conjectured that the models are integrable and provided conjectural expressions for the $S$ matrices, for both values of $\vartheta$ (see also [67]).

Let us now review some useful results of [30]. We start by reminding some basic facts about coset models based on symmetric spaces $G/H$. Their action can be written as

$$S = \frac{1}{g_0^2} \int \mathrm{d}^2 x \, \mathrm{Tr}\left(\partial_\mu U^{-1}(x) \partial_\mu U(x)\right),$$ (108)

where $U$ is a matrix field taking values in the Lie group $G$ and satisfying some additional constraints. In the case of the $SU(N)/SO(N)$ coset model, $U$ is a symmetric $SU(N)$ matrix. In the case of the $O(2P)/O(P) \times O(P)$ coset model, $U$ is a symmetric and traceless $O(2P)$ matrix. In the presence of a chemical potential $h$ coupled to a conserved charge $Q$, the action is modified as [42]

$$S = \frac{1}{g_0^2} \int \mathrm{d}^2 x \, \mathrm{Tr}\left(\overline{D}_\mu U^{-1}(x) D_\mu U(x)\right),$$ (109)

where

$$\begin{aligned} D_\mu U &= \partial_\mu U - h \delta_{\mu 0}(QU + UQ), \\ \overline{D}_\mu U^{-1} &= \partial_\mu U^{-1} + h \delta_{\mu 0}(U^{-1}Q + QU^{-1}), \end{aligned}$$ (110)

and the charge $Q$ is an element in the Cartan subalgebra $\mathfrak{h}$ of the Lie algebra $\mathfrak{g}$ of $G$. It is convenient to represent $Q$ by its dual, which is a vector $\boldsymbol{q} \in \Lambda_w \otimes \mathbb{R}$. Here, $\Lambda_w$ is the weight lattice. One can then calculate the free energy $\mathcal{F}(h)$ at one loop and obtain [42, 47]

$$\mathcal{F}(h) = -\frac{4h^2}{g^2(h)} \boldsymbol{q}^2 - \frac{h^2}{2\pi} \sum_{\boldsymbol{\alpha} \in \Delta_+} (\boldsymbol{q} \cdot \boldsymbol{\alpha})^2 \left\{ \log\left(|\boldsymbol{q} \cdot \boldsymbol{\alpha}|\right) - \frac{1}{2} \right\} + \cdots,$$ (111)

where $\Delta_+$ is the set of positive roots of $\mathfrak{g}$ and $g^2(h)$ is the running coupling constant at the scale set by $h$.

In [30], Fendley considered the free energy $\mathcal{F}(h)$ for the two coset models $SU(N)/SO(N)$ and $O(2P)/O(P) \times O(P)$, with particular choices of the charge $\boldsymbol{q}$. By comparing the perturbative result (111) to the Bethe Ansatz calculation of $\mathcal{F}(h)$, he checked his proposals for the $S$ matrices of the theory. Let us first consider the results for the $SU(N)/SO(N)$ coset sigma model. This model is asymptotically free, and in the convention (23) for the beta function, we have

$$\beta_0 = \frac{1}{8\pi\Delta}, \qquad \beta_1 = \frac{1}{128\pi^2} \frac{1 + 2\Delta}{\Delta^2},$$ (112)

where

$$\Delta = \frac{1}{N}.$$ (113)

We now consider the coupling to a conserved charge. When $G = SU(N)$, the charge $\boldsymbol{q}$ can be written as a vector in $\mathbb{R}^N$,

$$\boldsymbol{q} = (q_1, \cdots, q_N), \qquad \sum_{j=1}^n q_j = 0,$$ (114)

and the free energy (111) is then given by

$$\mathcal{F}(h) = -\frac{4h^2}{g^2(h)} \sum_{j=1}^N q_j^2 - \frac{h^2}{2\pi} \sum_{i<j} (q_i - q_j)^2 \left\{ \log\left(|q_i - q_j|\right) - \frac{1}{2} \right\} + \cdots$$ (115)

In [30], the following choice is made for the charge,

$$\boldsymbol{q} = \mathsf{r}(1, -1, 0, \cdots, 0), \tag{116}$$

where r is a normalization factor which is chosen to match the Bethe Ansatz calculation. Let us note that with this choice of charge there are two different types of particles in the ground state. By using the exact $S$-matrices proposed in [30], one finds that the free energy can be calculated from the integral equation (4) with the kernel

$$1 - \widetilde{K}(\omega) = e^{-\pi\Delta|\omega|} \frac{2\cosh\left(\frac{1}{2}(1 - 2\Delta)\pi\omega\right)\sinh(2\pi\Delta|\omega|)}{\cosh\left(\frac{1}{2}\pi\omega\right)}. \tag{117}$$

The Wiener–Hopf factorization of this kernel is given by[7]

$$G_+(\omega) = \frac{\exp\left(i\omega\left(-\Delta + 2\Delta\log(2\Delta) + \left(\frac{1}{2} - \Delta\right)\log\left(\frac{1}{2} - \Delta\right) + \frac{\log(2)}{2}\right) + i\Delta\omega\log(-i\omega)\right)}{2\sqrt{-\pi i\Delta\omega}}$$
$$\times \frac{\Gamma(1 - 2i\Delta\omega)\Gamma\left(\frac{1}{2} - i\left(\frac{1}{2} - \Delta\right)\omega\right)}{\Gamma\left(\frac{1}{2} - \frac{i\omega}{2}\right)}. \tag{118}$$

The perturbative free energy can be computed from the Bethe Ansatz at one loop by using (21). Since there are two particles in the ground state, we have to include an additional factor of 2 in (21), and by comparing to (115) we find that the normalization of the charge in (116) is $\mathsf{r}^2 = 1/8$ (a similar normalization was found in [68]). Agreement of the two expressions requires that [43]

$$\frac{\beta_1}{\beta_0^2} = a + \frac{1}{2}, \tag{119}$$

which holds in this model since $a = \Delta$, and tests the correctness of the $S$-matrix Ansatz proposed in [30]. From these results one can obtain in addition the mass gap, which is given by

$$\frac{m}{\Lambda} = \sqrt{\pi} \frac{2^{3\Delta+2}e^{-\frac{1}{2}-\Delta}}{\Gamma(1-\Delta)\Gamma(1+2\Delta)}. \tag{120}$$

The $O(2P)/O(P) \times O(P)$ coset sigma model has the same values for the coefficients of the beta function given in (112), with the only difference that now

$$\Delta = \frac{1}{2(P-1)}. \tag{121}$$

In [30] the following charge is considered

$$\boldsymbol{q} = \mathsf{r}\boldsymbol{\lambda}_1, \tag{122}$$

where $\boldsymbol{\lambda}_1$ is the highest weight of the fundamental representation of $O(2P)$, and r is a normalization constant. The ground state is then populated by a single species of particles, see [30] for details. For this choice of the charge, the kernel appearing in the Bethe Ansatz has the same form than (117), where $\Delta$ is now given in (121). This makes it possible to analyze both models simultaneously. By matching the perturbative and the Bethe Ansatz answers for the free energy one checks (119). We find $\mathsf{r}^2 = 1/8$, as well as the following value for the mass gap,

$$\frac{m}{\Lambda} = \sqrt{\pi} \frac{2^{5\Delta+2}e^{-\frac{1}{2}-\Delta}}{\Gamma(1-\Delta)\Gamma(1+2\Delta)}. \tag{123}$$

---

[7]This corrects some minor errors in the corresponding equation in [30].

## 4.2 The models at $\vartheta = 0$

At $\vartheta = 0$, the integral equations for these models are structurally identical to the "bosonic models" studied in [9, 19] (which include the $O(N)$ sigma model, see also [58]), and so we can easily apply the perturbative and non-perturbative analysis therein. From this section on, we will consider the observables as defined per excited particle species. This changes nothing in the case of $O(2P)/O(P) \times O(P)$, but in the case of $SU(N)/SO(N)$ there are two particle types in the ground state that obey the same Bethe Ansatz equation and so the total free energy is twice the free energy per particle, which we present, and so forth. To express these results, we introduce a coupling $\alpha$, analogous to (43),

$$\frac{1}{\alpha} + (\xi - 1)\log(\alpha) = \log\left(\frac{C\rho}{m}\right), \tag{124}$$

where

$$\xi = \frac{\beta_1}{\beta_0^2}, \qquad C = \frac{m}{\Lambda}\frac{1}{\mathfrak{c}\beta_0}, \tag{125}$$

using the convention (23) (which is distinct from the convention used in [9, 19]). In both families of coset models, we choose the arbitrary constant $\mathfrak{c}$ such that

$$C = \frac{\pi^{3/2} 2^{5\Delta + \frac{7}{2}} e^{-\Delta - 1/2}\Delta}{\Gamma(1 - \Delta)\Gamma(2\Delta + 1)}, \tag{126}$$

since with this choice the perturbative coefficients take the simplest possible form.

Using (118), we can apply the generalised Volin's method from [7, 9] to obtain the perturbative expansion of the ground state energy to high order. Following the procedure for "bosonic models", we find

$$\begin{aligned}
\frac{e}{\rho^2} = 4\pi\Delta\alpha\Bigg[ 1 &+ \frac{\alpha}{2} + \alpha^2\left(\frac{1}{4} + \frac{\Delta}{2}\right) + \alpha^3\left(\frac{5}{16} - \frac{3}{32}(7\zeta(3) - 10)\Delta + \frac{1}{16}(21\zeta(3) + 8)\Delta^2\right. \\
&+ \frac{1}{4}(-3\zeta(3) - 1)\Delta^3\bigg) + \alpha^4\left(\frac{53}{96} + \left(\frac{197}{96} - \frac{63\zeta(3)}{64}\right)\Delta + \left(\frac{97}{48} - \frac{35\zeta(3)}{16}\right)\Delta^2\right. \\
&+ \left(\frac{115\zeta(3)}{16} + \frac{1}{8}\right)\Delta^3 + \left(-\frac{19\zeta(3)}{4} - \frac{1}{4}\right)\Delta^4\bigg) + \mathcal{O}(\alpha^5)\Bigg].
\end{aligned} \tag{127}$$

An interesting property of this series is that the leading part of the coefficients in the large $N$ limit, i.e. $\Delta \to 0$, is identical to the leading part of the same coefficients in the large $N$ limit of the PCF model, see [6, 19]. We also note that the special case $\Delta = 1/2$ reduces to the $O(3)$ sigma model, up to factors of 2. However, the subsequent formulae do not apply to that case.

Let us now consider the non-perturbative corrections. According to the general results of [19], the trans-series for the normalized ground state energy is given by the formula

$$\frac{e}{\rho^2} = \frac{\alpha}{k^2}\left\{\varphi_0(\alpha) + \mathcal{C}_0^{\pm}e^{-\frac{2}{\alpha}}\alpha^{1-2\xi} + \mathcal{C}_1^{\pm}\left(e^{-\frac{2}{\alpha}}\alpha^{1-2\xi}\right)^{\xi_1}\varphi_1(\alpha) + \cdots\right\}, \tag{128}$$

where $\xi_1$ is the position of the first pole of $G_-(i\xi)$ and $\varphi_i(\alpha)$ are asymptotic series such that $\varphi_i(\alpha) = 1 + \mathcal{O}(\alpha)$. Using the definition (124), the constants are

$$\mathcal{C}_0^{\pm} = -\frac{C^2 G_+(i)k^2}{4\pi}iG'_-(i \pm 0), \tag{129}$$

$$\mathcal{C}_1^{\pm} = \frac{2i\sigma_1^{\pm}}{(\xi_1 - 1)^2\xi_1}\left(\frac{C^2 G_+^2(i)k^2}{8\pi}\right)^{\xi_1}, \tag{130}$$

where $\sigma_1^\pm$ is the residue of $\sigma(\omega)$ at the pole $\xi_1$. The free energy can be similarly written as

$$\mathcal{F}(h) = -\frac{k^2 h^2}{4\tilde{\alpha}} \left\{ \tilde{\varphi}_0(\tilde{\alpha}) - \mathrm{i}\sigma_1^\pm \left( \mathrm{e}^{-\frac{2}{\tilde{\alpha}}} \tilde{\alpha}^{1-2\xi} \right)^{\xi_1} \frac{2}{(\xi_1 - 1)^2 \xi_1} \left( \frac{G_+^2(\mathrm{i})}{2\pi k^2} \left( \frac{m}{\Lambda} \right)^2 \right)^{\xi_1} \tilde{\varphi}_1(\tilde{\alpha}) + \cdots \right\}$$
$$- \frac{m^2}{4\pi} \mathrm{i} G_+(\mathrm{i}) G'_-(\mathrm{i} \pm 0),$$
(131)

where $\tilde{\varphi}_i(\tilde{\alpha}) = 1 + \mathcal{O}(\tilde{\alpha})$, and $\tilde{\alpha}$ is given by

$$\frac{1}{\tilde{\alpha}} + \xi \log \tilde{\alpha} = \log \left( \frac{h}{\Lambda} \right).$$
(132)

We can now specialize these general formulae to the Fendley's models. In these models, the poles of $\sigma(\omega)$ for any $\Delta$ are located at

$$\frac{\ell}{1 - 2\Delta}, \quad \frac{\ell'}{2\Delta}, \quad \ell, \ell' \in \mathbb{N}.$$
(133)

The residues of $\sigma(\omega)$ associated to the first family formally contribute as terms with both real and (ambiguous) imaginary parts, while the second are purely real. For $\Delta < 1/4$ the leading pole occurs at

$$\xi_1 = \frac{1}{1 - 2\Delta},$$
(134)

and consequently the coefficients (129), (130) read

$$\mathcal{C}_0^\pm(0) = \left[ \sin(\pi\Delta) \mp \mathrm{i}\cos(\pi\Delta) \right] \left( \frac{64}{\mathrm{e}^2} \right)^\Delta \frac{\Gamma\left(\frac{1}{2} - \Delta\right)}{(\mathrm{e}\Delta)\Gamma\left(\Delta + \frac{1}{2}\right)},$$
$$\mathcal{C}_1^\pm(0) = \left[ -\sin\left( \frac{\pi\Delta}{1 - 2\Delta} \right) \pm \mathrm{i}\cos\left( \frac{\pi\Delta}{1 - 2\Delta} \right) \right] \left( \frac{1024^\Delta}{4\mathrm{e}^2(1 - 2\Delta)} \right)^{\frac{1}{1 - 2\Delta}} \frac{2\mathrm{e}(1 - 2\Delta)\Gamma\left(\frac{1 - 4\Delta}{2 - 4\Delta}\right)}{\Delta\Gamma\left(\frac{3 - 4\Delta}{2 - 4\Delta}\right)}.$$
(135)

The argument in the l.h.s. is to remind that these are the coefficients at $\vartheta = 0$. By using the asymptotic behavior of (127) and the numerical solution of the integral equation (9), we have tested the real and imaginary parts of $\mathcal{C}_{0,1}^\pm$ with great precision for several values of $\Delta$, as in [19]. The cases of $\Delta = 1/3, 1/4$ are conceptually similar but have some technical peculiarities, although formulae (129) and (130) still apply.

Let us finally note that, following [19, 31, 53], the real part of the $h$-independent term in (131) should give the free energy at $h = 0$, and we find

$$F(0) = -\frac{m^2}{16\sin(2\pi\Delta)}.$$
(136)

## 4.3 The models at $\vartheta = \pi$

Since these models remain integrable in the presence of the $\vartheta = \pi$ angle, one can analyse their Bethe Ansatze, similar to what we did for the $O(3)$ sigma model. As we will show, the resulting picture is also the same: the perturbative part does not change, non-perturbative formally imaginary ambiguous terms are also unchanged, but the non-perturbative formally real terms change sign. However, this change is organized differently. In the $O(3)$ sigma model, the residues $\mathrm{i}\sigma_n^\pm$ were purely imaginary and the real non-perturbative terms came from explicit driving terms. In Fendley's sigma models, the latter do not appear, but the residues have both a real and imaginary part. We will thus first show that, when compared to the $\vartheta = 0$ case,

only the residues change, and then we will show that this change is only in the sign of the real part.

Up to equation (71), the discussion in section 3.2 also applies to Fendley's coset models. In this case, the kernels are given by [30]

$$\phi_1(\omega) = \frac{1 - e^{2\pi\Delta|\omega|}}{1 + e^{\pi|\omega|}}, \qquad \phi_2(\omega) = \frac{1 + e^{\pi(1-2\Delta)|\omega|}}{1 + e^{\pi|\omega|}}, \tag{137}$$

and we easily verify (74), where now $\tilde{K}(\omega)$ is the kernel in (117) for the Fendley's coset models at $\vartheta = 0$. However, we no longer have the simplification of (75) that we found for $O(3)$. Instead, we find

$$\frac{\phi_2(\omega)}{1 - \phi_1(\omega)} = e^{-2\pi\Delta|\omega|}. \tag{138}$$

In order to extend this function to the complex plane, we define

$$\frac{\phi_2(\omega)}{1 - \phi_1(\omega)} = 1 - \frac{1}{d_+(\omega)d_-(\omega)}, \tag{139}$$

where

$$d_+(\omega) = \frac{e^{-i\Delta\omega[1-\log(-i\Delta\omega)]}}{\sqrt{-2\pi i\Delta\omega}} \Gamma(1 - i\Delta\omega), \tag{140}$$

and $d_-(\omega) = d_+(-\omega)$. An important ingredient for the analysis of the non-perturbative corrections at $\vartheta = \pi$ is the following result:

$$\frac{G_-(i\xi)}{d_+(i\xi)d_-(i\xi)} = 2\,\mathrm{Re}\big(G_-(i\xi)\big), \qquad \xi > 0. \tag{141}$$

This relation is easy to prove by observing that

$$d_+(i\xi \pm 0)d_-(i\xi \pm 0) = \frac{e^{\pm i\pi\left(\Delta\xi - \frac{1}{2}\right)}}{2\sin\left(\pi\Delta\xi\right)}, \qquad \xi > 0. \tag{142}$$

At the same time, we can write (118) as

$$G_-(i\xi \pm 0) = r(\xi)e^{\pm i\pi\left(\Delta\xi - \frac{1}{2}\right)}, \tag{143}$$

where $r(\xi)$ is a real function. Then it follows that

$$\frac{G_-(i\xi)}{d_+(i\xi)d_-(i\xi)} = 2\sin\left(\pi\Delta\xi\right)r(\xi), \qquad \xi > 0, \tag{144}$$

from where we conclude the result of (141).

Following a similar script to section 3.2, we take the $(+)$-projection of (71),

$$v = -\left[e^{-2iB\omega}\left\{\sigma\frac{\phi_2}{1-\phi_1}v\right\}(-\omega)\right]_+ + \left[\frac{ihG_+}{\omega - i0}\left(1 - e^{-2iB\omega}\frac{\phi_2}{1-\phi_1}\right)\right]_+ \\ - \frac{ime^B}{2}\left[G_+\left(\frac{1}{\omega - i} - \frac{\phi_2}{1-\phi_1}\frac{e^{-2iB\omega}}{\omega + i}\right)\right]_+. \tag{145}$$

We focus first on the integrals with $v$ in (145). The function

$$\sigma(\omega)\frac{\phi_2(\omega)}{1 - \phi_1(\omega)} = \sigma(\omega) - \frac{G_-(\omega)}{G_+(\omega)d_+(\omega)d_-(\omega)}, \tag{146}$$

is the analogue of (80) in the manipulations of the $O(3)$ case. Now, the extra term in (146) does have singularities in the upper half plane. However, it still has no branch cut, given that the ambiguities cancel in (141). In addition, the singularities of the extra term are located at $\mathrm{i}\xi_n$, the same positions as $\sigma(\omega)$, with residues $\tilde{\sigma}_n$. Since this function is not discontinuous, these residues are formally unambiguous. We can then deform the contour around the positive imaginary axis as usual, picking up the discontinuity of $\sigma(\omega)$ and the residues of both terms in (146):

$$\left[\mathrm{e}^{-2\mathrm{i}B\omega}\left\{\sigma\frac{\phi_2}{1-\phi_1}v\right\}(-\omega)\right]_+ = \frac{1}{2\pi\mathrm{i}}\int_{\mathcal{C}_\pm}\frac{\mathrm{e}^{-2B\xi'}\delta\sigma(\mathrm{i}\xi')v(\mathrm{i}\xi')}{\xi'-\mathrm{i}\omega}\mathrm{d}\xi' - \sum_{n\geq 1}\frac{\mathrm{e}^{-2B\xi_n}(\mathrm{i}\sigma_n^\pm - \mathrm{i}\tilde{\sigma}_n)v(\mathrm{i}\xi_n)}{\xi_n - \mathrm{i}\omega}. \tag{147}$$

As for the driving terms, similar manipulations give the terms proportional to $m$ as

$$\begin{aligned}\left[G_+\left(g_-^m + \mathrm{e}^{-2\mathrm{i}B\omega}\frac{\phi_2}{1-\phi_1}g_+^m\right)\right]_+ &= -\frac{\mathrm{i}m\mathrm{e}^B}{2}\frac{1}{2\pi\mathrm{i}}\int_{\mathcal{C}_\pm}\frac{\mathrm{e}^{-2B\xi'}\delta G_-(\mathrm{i}\xi')}{(\mathrm{i}\xi'+\omega)(\xi'-1)}\mathrm{d}\xi' \\ &\quad + \frac{\mathrm{i}m\mathrm{e}^B}{2}\frac{G_+(\omega)-G_+(\mathrm{i})}{\omega-\mathrm{i}} \\ &\quad + \frac{\mathrm{i}m\mathrm{e}^B}{2}\sum_{n\geq 1}\frac{\mathrm{e}^{-2B\xi_n}G_+(\mathrm{i}\xi_n)(\mathrm{i}\sigma_n^\pm - \mathrm{i}\tilde{\sigma}_n)}{(\mathrm{i}\xi_n+\omega)(\xi_n-1)},\end{aligned} \tag{148}$$

and, with some additional work,[8] the terms proportional to $h$ can also be written as

$$\begin{aligned}\left[\frac{\mathrm{i}hG_+}{\omega-\mathrm{i}0}\left(1-\mathrm{e}^{-2\mathrm{i}B\omega}\frac{\phi_2}{1-\phi_1}\right)\right]_+ &= \mathrm{i}h\frac{G_+(\omega)}{\omega} - \frac{\mathrm{i}h}{2\pi\mathrm{i}}\int_{\mathcal{C}_\pm}\frac{\mathrm{e}^{-2B\xi'}\delta G_-(\mathrm{i}\xi')}{\xi'(\mathrm{i}\xi'+\omega)}\mathrm{d}\xi' \\ &\quad + \mathrm{i}h\sum_{n\geq 1}\frac{\mathrm{e}^{-2B\xi_n}G_+(\mathrm{i}\xi_n)(\mathrm{i}\sigma_n^\pm - \mathrm{i}\tilde{\sigma}_n)}{\xi_n(\mathrm{i}\xi_n+\omega)}.\end{aligned} \tag{149}$$

The $\vartheta = 0$ case is recovered by removing the terms with $\mathrm{i}\tilde{\sigma}_n$. Thus, we can transform the $\vartheta = 0$ equation for $Q$ into the $\vartheta = \pi$ equation for $v$ simply by swapping $\{Q, \mathrm{i}\sigma_n^\pm\} \leftrightarrow \{v, \mathrm{i}\sigma_n^\pm - \mathrm{i}\tilde{\sigma}_n\}$.

The equation for $\epsilon_-$, which follows from $(-)$-projection of (71), is

$$\begin{aligned}\frac{\epsilon_-(\omega)}{G_-(\omega)} &= \left[\mathrm{e}^{-2\mathrm{i}B\omega}\{\sigma v\}(-\omega)\right]_- + \left[G_+\left(g_-^m + \mathrm{e}^{-2\mathrm{i}B\omega}g_+^m\right)\right]_- - \mathrm{i}h\left[G_+\frac{1-\mathrm{e}^{-2\mathrm{i}B\omega}}{\omega-\mathrm{i}0}\right]_- \\ &\quad - \left[\mathrm{e}^{-2\mathrm{i}B\omega}\left\{\frac{G_-v}{G_+d_+d_-}\right\}(-\omega)\right]_- - \left[\mathrm{e}^{-2\mathrm{i}B\omega}\left\{\frac{G_-(G_+g_-^m)}{G_+d_+d_-}\right\}(-\omega)\right]_- \\ &\quad - \left[\mathrm{e}^{-2\mathrm{i}B\omega}\frac{G_+}{G_-d_+d_-}\frac{\mathrm{i}hG_-}{\omega-\mathrm{i}0}\right]_-.\end{aligned} \tag{150}$$

The terms in the first line are identical to the $\vartheta = 0$ case, while those in the second and third are particular to $\vartheta = \pi$. We can write them in the generic form

$$\left[\mathrm{e}^{-2\mathrm{i}B\omega}\left\{\frac{G_-f}{G_+d_+d_-}\right\}(-\omega)\right]_- = \frac{1}{2\pi\mathrm{i}}\int_{\mathbb{R}}\frac{\mathrm{e}^{2\mathrm{i}B\omega'}G_-(\omega')}{G_+(\omega')d_+(\omega')d_-(\omega')}\frac{f(\omega)}{\omega'+\omega+\mathrm{i}0}\mathrm{d}\omega', \tag{151}$$

where

$$f(\omega) = v(\omega) - \mathrm{i}h\frac{G_+(\omega)}{\omega+\mathrm{i}0} + \frac{\mathrm{i}m\mathrm{e}^B}{2}\frac{G_+(\omega)}{\omega-\mathrm{i}}, \tag{152}$$

---

[8]The integral with the discontinuity is naively divergent due to the $1/\omega$ factor, however this can be fixed with the mouse-hole contour prescription described in Appendix A of [19].

is a function analytic in the upper half plane except at $\omega = i$. Since $G_-(i) = 0$, we can directly write such term for $\omega = -i\kappa$, with $\kappa \gg 1$, as a sum over residues, by deforming the contour around the positive real axis,

$$\frac{1}{2\pi i}\int_{\mathbb{R}}\frac{e^{2iB\omega'}G_-(\omega')}{G_+(\omega')d_+(\omega')d_-(\omega')}\frac{f(\omega)}{\omega'-i\kappa}d\omega' = -\sum_{n\geq 1}\frac{e^{-2B\xi_n}i\tilde{\sigma}_n f(i\xi_n)}{\xi_n - \kappa} + \mathcal{O}\left(e^{-2B\kappa}\right). \quad (153)$$

Plugging this result back into (150), we find

$$\begin{aligned}
\frac{\epsilon_+(i\kappa)}{G_+(i\kappa)} = &\frac{1}{2\pi i}\int_{\mathcal{C}_\pm}\frac{e^{-2B\xi}\delta\sigma(i\xi)\nu(i\xi)}{\xi - \kappa}d\xi - \frac{me^B}{2}\frac{G_+(i)}{1+\kappa}\\
&+ \frac{me^B}{2}\frac{1}{2\pi i}\int_{\mathcal{C}_\pm}\frac{e^{-2B\xi}G_+(i\xi)\delta\sigma(i\xi)}{(\xi-1)(\xi-\kappa)}d\xi - \frac{h}{2\pi i}\int_{\mathcal{C}_\pm}\frac{e^{-2B\xi}G_+(i\xi)\delta\sigma(i\xi)}{\xi(\xi-\kappa)}d\xi\\
&-\sum_{n\geq 1}\frac{e^{-2B\xi_n}}{\xi_n - \kappa}(i\sigma_n^\pm - i\tilde{\sigma}_n)\left(\nu(i\xi_n) - \frac{hG_+(i\xi_n)}{\xi_n} + \frac{me^B G_+(i\xi_n)}{2(\xi_n - 1)}\right) + \mathcal{O}\left(e^{-2B\kappa}\right). \quad (154)
\end{aligned}$$

We can see that in the boundary condition imposed by $\lim_{\kappa\to\infty}\kappa\epsilon_+(i\kappa)$ the only difference between the $\vartheta = 0$ and $\vartheta = \pi$ cases are the residues, since $i\tilde{\sigma}_n$ are absent in the former case. The remaining terms, which are identical in both, do not contain further non-perturbative corrections, other than those that come through $\nu$ from (145).

As we saw in section 3.2, to calculate the free energy we need also to evaluate (150) at $\omega = -i$. This differs from (154) because there is now an additional pole at $\omega = i$ when deforming some of the integrals. In the end we find

$$\begin{aligned}
\frac{\epsilon_+(i)}{G_+(i)} = &\frac{1}{2\pi i}\int_{\mathcal{C}_\pm}\frac{\delta\sigma(i\xi)}{\xi - 1}\left(e^{-2B\xi}\nu(i\xi) + \frac{me^B e^{-2B\xi}G_+(i\xi)}{2(\xi-1)} - \frac{hG_+(i\xi)e^{-2B\xi}}{\xi}\right)d\xi\\
&-\frac{me^B}{4}G_+(i) - \sum_{n\geq 1}\frac{e^{-2B\xi_n}(i\sigma_n^\pm - i\tilde{\sigma}_n)}{\xi_n - 1}\left(\nu(i\xi_n) - \frac{hG_+(i\xi_n)}{\xi_n} + \frac{me^B G_+(i\xi_n)}{2(\xi_n - 1)}\right)\\
&+\frac{me^{-B}}{2}\left(iG_-'(i\pm 0) - 2\,\mathrm{Re}\left\{iG_-'(i)\right\}\right). \quad (155)
\end{aligned}$$

In the last term, we used (141). For $\vartheta = 0$, as follows from the analysis of [19], we have

$$\begin{aligned}
\frac{\epsilon_+(i)}{G_+(i)} = &\frac{1}{2\pi i}\int_{\mathcal{C}_\pm}\frac{\delta\sigma(i\xi)}{\xi - 1}\left(e^{-2B\xi}Q(i\xi) + \frac{me^B e^{-2B\xi}G_+(i\xi)}{2(\xi-1)} - \frac{hG_+(i\xi)e^{-2B\xi}}{\xi}\right)d\xi\\
&-\frac{me^B}{4}G_+(i) - \sum_{n\geq 1}\frac{e^{-2B\xi_n}i\sigma_n^\pm}{\xi_n - 1}\left(Q(i\xi_n) - \frac{hG_+(i\xi_n)}{\xi_n} + \frac{me^B G_+(i\xi_n)}{2(\xi_n - 1)}\right)\\
&+\frac{me^{-B}}{2}iG_-'(i\pm 0). \quad (156)
\end{aligned}$$

We can then see that the difference between these two equations is simply in the residues from the poles at $i\xi_n$ as well as in the so-called IR pole. Since the first term in the second line is perturbative ($me^B \sim h$), the difference is then restricted to the non-perturbative terms, as expected.

In (155), it is already manifest that the change to the IR pole is that the real part changes sign while the ambiguous imaginary part remains the same. In addition, as a consequence of (141), we also have

$$i\sigma_n^\pm - i\tilde{\sigma}_n = -\,\mathrm{Re}(i\sigma_n^\pm) + i\,\mathrm{Im}(i\sigma_n^\pm), \quad (157)$$

so the real part of the residues also changes sign.

Having found the map between the $\vartheta = 0$ trans-series solution and that of $\vartheta = \pi$, we can apply it to our results in section 4.2. The trans-series for the ground state energy is given by (128), but with the constants

$$
\mathcal{C}_0^\pm(\pi) = \left[-\sin(\pi\Delta) \mp i\cos(\pi\Delta)\right]\left(\frac{64}{e^2}\right)^\Delta \frac{\Gamma\left(\frac{1}{2}-\Delta\right)}{(e\Delta)\Gamma\left(\Delta+\frac{1}{2}\right)},
$$

$$
\mathcal{C}_1^\pm(\pi) = \left[\sin\left(\frac{\pi\Delta}{1-2\Delta}\right) \pm i\cos\left(\frac{\pi\Delta}{1-2\Delta}\right)\right]\left(\frac{1024^\Delta}{4e^2(1-2\Delta)}\right)^{\frac{1}{1-2\Delta}} \frac{2e(1-2\Delta)\Gamma\left(\frac{1-4\Delta}{2-4\Delta}\right)}{\Delta\Gamma\left(\frac{3-4\Delta}{2-4\Delta}\right)},
$$

(158)

while the free energy can be written as

$$
\begin{aligned}
\mathcal{F}(h,\pi) = &-\frac{k^2 h^2}{4\tilde{\alpha}}\left\{1 - 2\left(-\text{Re}\left\{i\sigma_1^+\right\} \pm i\,\text{Im}\left\{i\sigma_1^+\right\}\right)\left(e^{-\frac{2}{\tilde{\alpha}}}\tilde{\alpha}^{1-2\xi}\right)^{\xi_1}\frac{\left(\frac{G_+^2(i)}{2\pi k^2}\left(\frac{m}{\Lambda}\right)^2\right)^{\xi_1}}{(\xi_1-1)^2\,\xi_1} + \cdots\right\} \\
&-\frac{m^2}{4\pi}G_+(i)\left(-\text{Re}\left\{iG_-'(i)\right\} \pm i\,\text{Im}\left\{iG_-'(i+0)\right\}\right).
\end{aligned}
$$

(159)

In either of these representations, one can see that the leading, real exponentially suppressed effects change sign, including $F(0,\pi)$, which now reads

$$
F(0,\pi) = \frac{m^2}{16\sin(2\pi\Delta)}.
$$

(160)

Meanwhile, the formally imaginary terms, responsible for canceling the ambiguities induced by the large order behavior of the perturbative series, remain unchanged.

In the $O(3)$ sigma model we identified the effects that change sign as instantons, but in Fendley's coset sigma models there are both instantons and renormalons. We observe this in the first sequence of singularities in (133), which survives the large $N$ limit $\Delta \to 0$, while the second sequence does not (this is similar to what happens in the $O(N)$ supersymmetric non-linear sigma model studied in [19]). Following the analysis of [19], we identify the first sequence as "renormalon-like" and the second as "instanton-like". In fact, in (131) and (159) we found that the free energy has the leading exponential correction $e^{-2\xi_1/\tilde{\alpha}}$ due to the "renormalon-like" singularity $\xi_1 = 1/(1-2\Delta)$ of $\sigma(\omega)$, which is the closest singularity to the origin. The change in the real part of the residues applies equally to both types of singularities, i.e. the effect of the $\vartheta = \pi$ term on renormalon contributions is identical to its effect on instanton contributions. Hence, our results provide evidence that, in Fendley's coset sigma models, the real non-perturbative effects associated to renormalons change sign in the presence of a non-trivial theta angle $\vartheta = \pi$.

## 4.4 Testing the analytic results

As we did for the $O(3)$ sigma model, we now test the validity of the analytic expressions we obtained for the non-perturbative effects in Fendley's integrable models. In particular, we check numerically the real exponential corrections to the normalized energy density $e/\rho^2$, as presented in (128), with the coefficients of (135) (for $\vartheta = 0$) and (158) (for $\vartheta = \pi$). As in the case of the $O(3)$ sigma model, the numerical study of the integral equation (54) is subtle, and we used again the methods described in Appendix B.

In Fig. 5, we plot the points $(\alpha, k^2 e/\rho^2)$ that we obtained numerically from the Bethe Ansatz, by using $\Delta = 1/5$, at both $\vartheta = 0$ and $\vartheta = \pi$. We compare the result to the Borel resummation of the perturbative series, shown as the dashed line. When $\vartheta = \pi$, we used

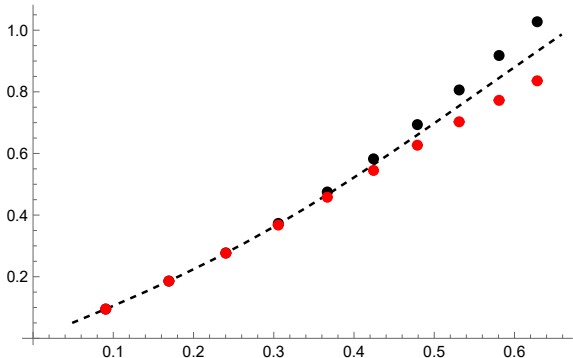

Figure 5: Numerical evaluation of $(\alpha, k^2 e/\rho^2)$ for Fendley's integrable model with $\Delta = 1/5$, at $\vartheta = 0$ (black dots) and $\vartheta = \pi$ (red dots). The dashed line corresponds to the Borel resummation of the perturbative part, $\alpha \operatorname{Re}\left[s_{\pm}(\varphi)(\alpha)\right]$, which is shared by both values of the angle $\vartheta$.

the values $B = 10/k$, with integers $1 \le k \le 10$. When $\vartheta = 0$, we looked for the values of $B$ that would result into the same numerical $\alpha$ that we found for $\vartheta = \pi$. To obtain the Borel resummation of the perturbative series, we used 81 coefficients, computed with the generalization of Volin's method.

In Fig. 6 and Table 2, we consider the difference between the numerical result for the Bethe Ansatz, and the Borel resummation of the perturbative series. We compare the result against the analytic prediction of (128), and we find good agreement for $\Delta = 1/3, 1/5, 1/6$, at both $\vartheta = 0$ and $\vartheta = \pi$. The leading exponential effect due to the IR pole, represented by the dashed line, changes sign. This confirms the change of sign of the $F(0)$ term, i.e. the $h$-independent term in the free energy. As for the subleading exponential effect, for $\Delta = 1/5$ and $1/6$ it corresponds to a renormalon of the type first identified in [19]. The numerical test confirms that the real part of its contribution does indeed change sign as we go from $\vartheta = 0$ to $\vartheta = \pi$, as can be seen from the solid lines. On the other hand, for $\Delta = 1/3$, the singularity at $\xi_1$ corresponds to an instanton, and we confirm that the associated contribution also changes sign, as expected. In this case, we mention that instead of the coefficient $\mathcal{C}_1^{\pm}$ in (135) and (158), which is only valid for the renormalon singularities, we had to use the the more general coefficient in (130) with $\xi_1 = 3$ and its corresponding $\vartheta = \pi$ counterpart.

## 5 Discussion and prospects for future work

In this paper we continued our analytic study of non-perturbative corrections in integrable field theories, which we started in [19]. We considered integrable sigma models that admit a theta angle, and we studied the effect of this angle on exponentially small corrections due to renormalons and instantons. As anticipated in the study of the deformed sigma models in [30, 31], we found that would-be instanton corrections change sign as we turn on $\vartheta = \pi$. We also found that, in some cases, the behavior of renormalon corrections is more subtle: the imaginary parts of the residues associated to renormalon singularities do not change sign, but their real parts do.[9] This leads to renormalon contributions which are schematically of the form,

$$(\mathcal{C}_1(\vartheta) + i\mathcal{C}_2)g^b e^{-A/g^2}\Phi(g), \tag{161}$$

---

[9]In reaching this conclusion, we have assumed a renormalon-like nature for these contributions, based for example on the fact that they survive the large $N$ limit.

where $\Phi(g)$ is a formal power series and $b$ is an appropriate exponent. Most of the ingredients of this trans-series, i.e. $A$, $b$ and $\Phi(g)$, can be in principle obtained from the large order behavior of the perturbative series, through its imaginary part, proportional to $\mathcal{C}_2$. However, the trans-series parameter $\mathcal{C}_1(\vartheta)$, which is sensitive to the theta angle, cannot be predicted from perturbation theory alone. Such a scenario for renormalon corrections was anticipated in e.g. [69], and the models discussed in this paper seem to provide a concrete realization thereof. The possibility that the overall coefficients associated to renormalon corrections might depend on the theta angle, as we are discovering here, was already pointed out in [70]. Additionally,

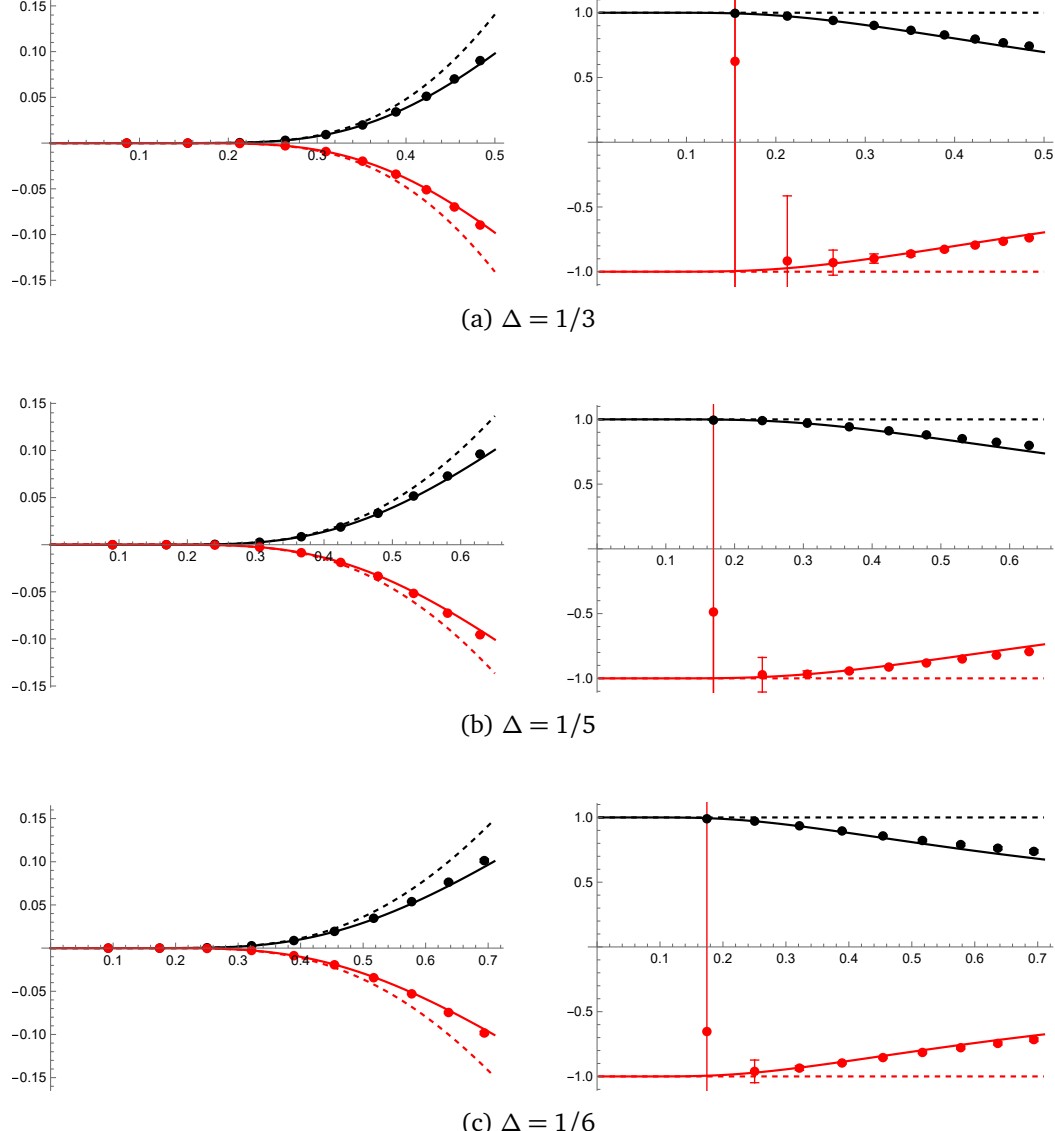

Figure 6: In the figures on the left we plot the difference between the normalized energy density $k^2 e/\rho^2$ and the real part of the Borel resummation of the perturbative series, for the Fendley's integrable models at $\vartheta = 0$ (black) and $\vartheta = \pi$ (red). The dots are the numerical calculations of the difference. The line is the theoretical prediction, given by the exponential corrections of (128), including the IR renormalon correction of order $e^{-2/\alpha}$ (dashed line) and also the next subleading correction, of order $e^{-2\xi_1/\alpha}$ (full line). In the figures on the right, the result has been normalized by dividing by the IR renormalon correction.

Table 2: Difference between the normalized energy density $k^2 e/\rho^2$ and the real part of the Borel resummation of the perturbative series, for the Fendley's integrable model with $\Delta = 1/5$. We compare the numerical value obtained from the Bethe Ansatz (2nd and 3rd column), to the exponential corrections of (128) (4th column). The error estimates of $k^2 e/\rho^2$ are found by varying the quadrature parameters, see Appendix B for the details of the $\vartheta = \pi$ case.

| $\alpha$ | Numerical ($\vartheta = 0$) | Numerical ($\vartheta = \pi$) | Asymptotics |
|---|---|---|---|
| 0.090553 | $(0 \pm 6) \times 10^{-7}$ | $(3 \pm 18) \times 10^{-6}$ | $\pm 2.32356 \times 10^{-10}$ |
| 0.169263 | $9.75(28 \pm 32) \times 10^{-6}$ | $-0.0000(05 \pm 35)$ | $\pm 9.75205 \times 10^{-6}$ |
| 0.240275 | $0.000392(06 \pm 28)$ | $-0.0003(8 \pm 5)$ | $\pm 0.000392$ |
| 0.305661 | $0.00263(31 \pm 31)$ | $-0.0026(3 \pm 7)$ | $\pm 0.002635$ |
| 0.366721 | $0.0084(80 \pm 14)$ | $-0.0084(8 \pm 9)$ | $\pm 0.008496$ |
| 0.424307 | $0.0187(7 \pm 4)$ | $-0.018(79 \pm 11)$ | $\pm 0.018811$ |
| 0.478970 | $0.0333(8 \pm 9)$ | $-0.033(41 \pm 15)$ | $\pm 0.033460$ |
| 0.531058 | $0.051(69 \pm 15)$ | $-0.051(65 \pm 20)$ | $\pm 0.051775$ |
| 0.580792 | $0.072(90 \pm 24)$ | $-0.072(64 \pm 28)$ | $\pm 0.072905$ |
| 0.628315 | $0.096(19 \pm 34)$ | $-0.095(5 \pm 4)$ | $\pm 0.096016$ |

the effect of the theta angle is not an overall multiplicative factor, in contrast with the instanton case. It is not obvious how this can be implemented in an hypothetical saddle-point construction of renormalons, where the theta term in the action must appear as overall phases. More work is needed however to provide a microscopic description of the non-perturbative corrections unveiled in these coset sigma models, and to verify the instanton/renormalon nature of the non-perturbative corrections.

Our analysis also establishes that purely real corrections which change sign as we change the theta angle can not be detected in any way by a resurgent analysis of the perturbative series. These include notably the would-be instanton correction of the $O(3)$ sigma model already discussed in [19, 24]. We conclude that, in the terminology of [6, 19], these theories satisfy the weak version of the resurgence program (physical observables are given by Borel-resummed trans-series), but not the strong version, since not all the trans-series corrections can be decoded from the perturbative series alone. This is however expected in theories which admit a theta angle (see [17] for a similar discussion).

As we explained in the introduction, one important motivation for this work was to obtain exact results on non-perturbative effects in semi-realistic asymptotically free theories, which can serve as signposts for non-perturbative methods in quantum field theory. In the case of renormalon corrections, there is no known method to independently test the predictions we have obtained. However, the analytic result (41) provides a non-trivial prediction for an instanton correction in the $O(3)$ sigma model, which could and should be testable by conventional methods. As we pointed out in (102), this correction leads to a non-vanishing topological susceptibility in the presence of the $h$ field, and this prediction from integrability might be testable in a lattice calculation, along the lines of [32]. Instanton calculus in non-supersymmetric theories has been an "arena for bloody controversies", in the famous description by Sidney Coleman [71], and we hope that this clean prediction from integrability might help clarify some of the issues at stake. The presence of an external field $h$ should act as a regulator for the various IR and UV singularities that afflict instanton calculations in the $O(3)$ sigma model, but understanding in a precise way how this works is not obvious, at least for us.

## Acknowledgments

We would like to thank Ivan Graham and specially Martin Gander and Walter Gautschi for very useful advice on the numerical study of the integral equation (54). The numerical computations were performed at University of Geneva on "Baobab" and "Yggdrasil" HPC clusters.

**Funding information**    This work has been supported by the ERC-SyG project "Recursive and Exact New Quantum Theory" (ReNewQuantum), which received funding from the European Research Council (ERC) under the European Union's Horizon 2020 research and innovation program, grant agreement No. 810573. In the reviewing stage of this work, T.R. was supported by the ERC-COG grant NP-QFT No. 864583 "Non-perturbative dynamics of quantum fields: from new deconfined phases of matter to quantum black holes", by the MUR-FARE2020 grant No. R20E8NR3HX "The Emergence of Quantum Gravity from Strong Coupling Dynamics", and by INFN Iniziativa Specifica ST&FI.

## A    Calculation of non-perturbative corrections in the $O(3)$ sigma model

In this Appendix we give some details on the computation of the non-perturbative effects in $\mathcal{F}(h)$ for the $O(3)$ sigma model. The first step in this computation is to find the explicit exponential corrections arising from $Q_n = Q(i\xi_n)$ in (33). Since we want to go to order $e^{-4B}$ (second order in the exponential correction), we will need to compute $Q_1$ to order $e^{-2B}$, but only the perturbative part of $Q_2$.

Let us first compute $Q_1$. We evaluate (15) at $\xi = \xi_1$ and obtain the equation

$$Q_1 + \frac{1}{2\pi i}\int_0^\infty \frac{e^{-2B\xi}\delta\sigma(i\xi)Q(i\xi)}{\xi + \xi_1}d\xi + \frac{e^{-2B\xi_1}i\sigma_1^\pm Q_1}{2\xi_1} = \frac{1}{2\pi i}\int_\mathbb{R}\frac{G_-(\omega)g_+(\omega)}{\omega + i\xi_1}d\omega. \quad (A.1)$$

In the case of the $O(3)$ sigma model, we have $\sigma_1^\pm = 0$. Thus, we only have to consider the exponential corrections arising from the last integral. To do this calculation, we use the expansion (20), where the coefficients $k, a, b$ can be read from (26) and are given by

$$k = \frac{1}{\sqrt{\pi}}, \qquad a = \frac{1}{2}, \qquad b = \frac{\gamma_E - 1 - \log(2)}{2}. \quad (A.2)$$

The calculation of the integral can be done with appropriate contour deformations, similar to the ones in [19], and we eventually find

$$Q_1 = -kh(2B)^{1/2}\frac{\sqrt{\pi}}{2} + \frac{me^B}{2}\left[-iG_+'(i) + \frac{G_-(i)}{2}e^{-2B}\right], \quad (A.3)$$

at the order in exponentially small corrections that we are considering. The perturbative part of $Q_2$ is obtained by repeating the steps leading to (A.3), and we find

$$Q_2 = -kh(2B)^{1/2}\frac{\sqrt{\pi}}{4} - \frac{me^B}{2}\left[G_+(2i) - G_+(i)\right] + \mathcal{O}(B^0). \quad (A.4)$$

In the computation we will also need the values of the function $Q(i\xi)$ for "small" arguments $\xi = x/2B$. At the order we are working, and since $\sigma_1^\pm = 0$, we can ignore the $Q_n$ terms in (31), and we obtain

$$Q(ix/2B) + \frac{1}{2\pi i}\int_0^\infty \frac{e^{-y}\delta\sigma(iy/2B)Q(iy/2B)}{y + x}dy = \frac{1}{2\pi i}\int_\mathbb{R}\frac{G_-(\omega)g_+(\omega)}{\omega + ix/2B}d\omega. \quad (A.5)$$

To solve this equation we write a trans-series Ansatz for $Q(\mathrm{i}x/2B)$:

$$Q(\mathrm{i}x/2B) = Q_{(0)}(\mathrm{i}x/2B) + Q_{(1)}(\mathrm{i}x/2B)\mathrm{e}^{-2B} + \mathcal{O}(\mathrm{e}^{-4B}), \tag{A.6}$$

as well as for the driving term:

$$\frac{1}{2\pi\mathrm{i}} \int_{\mathbb{R}} \frac{G_-(\omega)g_+(\omega)}{\omega + \mathrm{i}x/2B} \mathrm{d}\omega = r_{(0)}(x) + r_{(1)}(x)\mathrm{e}^{-2B} + \mathcal{O}(\mathrm{e}^{-4B}). \tag{A.7}$$

The perturbative part $r_{(0)}(x)$ was computed at next-to-leading order in $1/B$ in [19] (where it was simply denoted by $r(x)$). The exponential correction in (A.7) arises from the pole at $\omega = \mathrm{i}$ in $g_+(\omega)$, and yields

$$r_{(1)}(x) = \frac{m\mathrm{e}^B}{2} \frac{G_-(\mathrm{i})}{1 + x/2B} = \frac{m\mathrm{e}^B}{2} G_-(\mathrm{i})\left[1 + \mathcal{O}(B^{-1})\right]. \tag{A.8}$$

To write the integral equations satisfied by the functions $Q_{(0)}(\mathrm{i}x/2B)$, $Q_{(1)}(\mathrm{i}x/2B)$, it is convenient to use the Airy integral operator introduced in [19], which is defined by

$$(\mathsf{K}f)(x) = \int_0^\infty \frac{\mathrm{e}^{-y}}{x + y} f(y)\mathrm{d}y. \tag{A.9}$$

The perturbative function $q(x) = Q_{(0)}(\mathrm{i}x/2B)$ solves [40, 41]

$$q(x) + \frac{1}{2\pi\mathrm{i}} \int_0^\infty \frac{\mathrm{e}^{-y}\delta\sigma(\mathrm{i}y/2B)}{x + y} q(y)\mathrm{d}y = r_{(0)}(x). \tag{A.10}$$

The explicit solution to this equation at leading order in $1/B$ is given by

$$Q_{(0)}(\mathrm{i}x/2B) = -kh(2B)^{1/2}\left[Bq_{(0),0}(x) + \mathcal{O}(B^0)\right], \tag{A.11}$$

where [19]

$$q_{(0),0}(x) = \frac{\mathrm{e}^{x/2}}{\sqrt{x}}\left(K_1\left(\frac{x}{2}\right) - \frac{2\mathrm{e}^{-x/2}}{x}\right), \tag{A.12}$$

and $K_\nu(x)$ is the modified Bessel function. By plugging the Ansatz (A.6) in (A.5), we find

$$Q_{(1)}(\mathrm{i}x/2B) = \frac{m\mathrm{e}^B}{2} G_-(\mathrm{i})\left[q_{(1),0}(x) + \mathcal{O}(B^{-1})\right], \tag{A.13}$$

where $q_{(1),0}(x)$ satisfies

$$\left(1 + \frac{\mathsf{K}}{\pi}\right)q_{(1),0} = 1. \tag{A.14}$$

Let us define, as in [19],

$$\langle f \rangle = \int_0^\infty \mathrm{e}^{-x} f(x)\mathrm{d}x. \tag{A.15}$$

In order to obtain the free energy, we do not need the solutions to the integral equations, but rather their "average" (A.15). We have [19]

$$\langle q_{(0),0} \rangle = \sqrt{\pi}(4 - \pi), \qquad \langle q_{(1),0} \rangle = \frac{\pi}{4}. \tag{A.16}$$

Let us now use the results above to compute $\epsilon_+(i)$ up to second order in the exponential correction. (33) is singular at $\kappa = 1$, but the correct expression can be obtained by doing the correct residue calculation at $\omega = i$ in (17), and we obtain

$$\frac{\epsilon_+(i)}{G_+(i)} = \frac{1}{2\pi i}\int_{\mathbb{R}} \frac{G_-(\omega)g_+(\omega)}{\omega - i}d\omega + \frac{1}{2\pi i}\int_0^\infty \frac{e^{-2B\xi}\delta\sigma(i\xi)Q(i\xi)}{\xi - 1}d\xi$$
$$+ e^{-2B}\sigma(i)Q_1 - i\sigma_2^\pm e^{-4B}Q_2. \quad \text{(A.17)}$$

The first term in the r.h.s. can be evaluated by residue calculus. The second term is an integral involving the discontinuity $\delta\sigma(i\xi)$ and it can be calculated by using the explicit result for $Q(ix/2B)$. Putting everything together, and using our expressions for $Q_1$ and $Q_2$, we obtain

$$\frac{\epsilon_+(i)}{G_+(i)} = \frac{kh(2B)^{1/2}}{\pi}\left[2\sqrt{\pi} - \frac{\langle q_{(0),0}\rangle}{2}\right] - \frac{me^B}{2}\frac{G_+(i)}{2} + \left[h\left(-k(2B)^{1/2}\frac{\sqrt{\pi}}{2}\frac{G_-(i)}{G_+(i)} - G_-(i)\right)\right.$$
$$+ \frac{me^B}{2}\left(-2BG_-(i) + iG_-'(i\pm 0) - iG_+'(i)\frac{G_-(i)}{G_+(i)}\right)\right]e^{-2B}$$
$$+ \left[h\left(k(2B)^{1/2}\frac{\sqrt{\pi}}{4}i\sigma_2^\pm + \frac{G_+(2i)}{2}i\sigma_2^\pm\right) + \frac{me^B}{2}\left(\frac{G_-^2(i)}{2G_+(i)} - i\sigma_2^\pm G_+(i)\right)\right]e^{-4B}. \quad \text{(A.18)}$$

The next ingredient is the boundary condition, relating $me^B$ with $h$. We have to calculate (18), including exponential corrections. We consider (33), and we write explicitly the second exponential correction (the first correction vanishes due to $\sigma_1^\pm = 0$):

$$\frac{\epsilon_+(i\kappa)}{G_+(i\kappa)} = \frac{1}{2\pi i}\int_{\mathbb{R}}\frac{G_-(\omega)g_+(\omega)}{\omega - i\kappa}d\omega + \frac{1}{2\pi i}\int_0^\infty \frac{e^{-2B\xi}\delta\sigma(i\xi)Q(i\xi)}{\xi - \kappa}d\xi - \frac{i\sigma_2^\pm e^{-2B\xi_2}}{\xi_2 - \kappa}Q_2. \quad \text{(A.19)}$$

In determining the boundary condition, we have to push the calculation up to order $1/B$. The perturbative part was evaluated in detail in [19]. The non-perturbative corrections can be calculated by using the results above. One finds,

$$\frac{\epsilon_+(i\kappa)}{G_+(i\kappa)} = \cdots + h\frac{G_+(i\xi_2)}{(\xi_2 - \kappa)\xi_2}i\sigma_2^\pm e^{-2B\xi_2}$$
$$+ \frac{me^B}{2}\left[\left(\frac{1}{1-\kappa} + \frac{1}{8B\kappa}\right)G_-(i)e^{-2B} - \frac{G_+(i\xi_2)}{(\xi_2 - \kappa)(\xi_2 - 1)}i\sigma_2^\pm e^{-2B\xi_2}\right], \quad \text{(A.20)}$$

where $\cdots$ indicates the perturbative part. Now we impose $\lim_{\kappa\to+\infty}\kappa\epsilon_+(i\kappa) = 0$ and, taking into account that $\lim_{\kappa\to+\infty}G_+(i\kappa) = 1$, we find

$$me^B = kh(2B)^{1/2}\frac{\sqrt{\pi}}{G_+(i)}\left[1 + \frac{\log(2B)c_{(0),1,1}}{B} + \frac{c_{(0),1,0}}{B}\right.$$
$$+ \left.\left(c_{(1),0} + \frac{\log(2B)c_{(1),1,1}}{B} + \frac{c_{(1),1,0}}{B}\right)e^{-2B\xi_1} + c_{(2),0}e^{-2B\xi_2}\right], \quad \text{(A.21)}$$

where the coefficients have the following values:

$$c_{(0),1,1} = \frac{1}{4}, \qquad\qquad c_{(0),1,0} = \frac{-1 + \log(64)}{8}, \qquad \text{(A.22)}$$

$$c_{(1),0} = -\frac{G_-(i)}{G_+(i)}, \qquad\qquad c_{(1),1,1} = -\frac{G_-(i)}{4G_+(i)}, \qquad \text{(A.23)}$$

$$c_{(1),1,0} = -\frac{G_-(i)}{G_+(i)}\frac{-3 + \log(64)}{8}, \qquad\qquad c_{(2),0} = \frac{G_-^2(i)}{G_+^2(i)} + \frac{i\sigma_2^\pm}{2}. \qquad \text{(A.24)}$$

The final step to construct the free energy from (19), using the expression for $\epsilon_+(i)$ we found in (A.18) and the boundary condition (A.21). This leads to the result of (35).

# B   Numerical analysis of the integral equations

In this Appendix, we explain how we solved numerically the integral equation (54), and computed $e$ and $\rho$, for both the $O(3)$ sigma model and Fendley's integrable models at $\vartheta = \pi$. Due to the unbounded integration interval $(-\infty, B]$, many technical complications arise that are not present in the integral equations for $\vartheta = 0$, leading to worse numerical precision for $e$ and $\rho$. We solved these issues with a combination of brute force and by using a specially adapted computation strategy.

Let us consider a general Fredholm integral equation of the second kind:

$$v(x) - \int_{-\infty}^{B} k(x,y)v(y)\mathrm{d}y = g(x). \tag{B.1}$$

Our goal is to solve numerically for $v(x)$ so as to find good approximations for the quantities

$$\rho = \frac{1}{4\pi}\int_{-\infty}^{B} v(x)\mathrm{d}x, \qquad e = \frac{1}{4\pi}\int_{-\infty}^{B} \mathrm{e}^{x} v(x)\mathrm{d}x. \tag{B.2}$$

To find a numerical approximation to $v$, we split the interval $(-\infty, B]$ with a grid of points $x_i$, $i = 1, \ldots, N$. Then the integral equation (B.1) can be approximated as

$$\sum_{j=1}^{N}\left(\delta_{ij} - K_{ij}\right)v(x_j) = g(x_i), \tag{B.3}$$

where $K$ is the quadrature matrix, defined by

$$K_{ij} = w_j k(x_i, x_j). \tag{B.4}$$

The weights $w_i$ will depend on the quadrature we use to approximate the integral. The discretized integral equation in (B.3) can then be solved for $v(x_i)$ with standard techniques, and we can approximate $\rho$ and $e$ as

$$\rho \approx \frac{1}{4\pi}\sum_{i=1}^{N} w_i v(x_i), \qquad e \approx \frac{1}{4\pi}\sum_{i=1}^{N} w_i \mathrm{e}^{x_i} v(x_i). \tag{B.5}$$

Quadratures for integral equations of the form (B.1) were discussed in e.g. [72, 73], but when applied to the $O(3)$ and the Fendley's models, their convergence to the correct results was too slow. We then consider the following composite quadrature.[10] We split the interval $(-\infty, B]$ in the subinterval $[-b, B]$ (with $b > 0$), where we apply a Gauss–Legendre quadrature with $n$ points, and the subinterval $(-\infty, -b]$, where we apply a Gauss–Jacobi quadrature with $m$ points. For a general integral in $(-\infty, B]$, this composite quadrature will yield the approximation

$$\int_{-\infty}^{B} f(x)\mathrm{d}x \approx \sum_{j=1}^{n+m} w_j f(x_j). \tag{B.6}$$

Let us compute the nodes $x_i$ and the associated weights $w_i$. In the interval $[-1, 1]$, the nodes $X_i$ for the Gauss–Legendre quadrature are given by the roots of the Legendre polynomial of degree $n$, $L_n$. The associated weights are then given by

$$W_i = \frac{2}{(1 - X_i^2)[L_n'(X_i)]^2}. \tag{B.7}$$

---

[10] We thank Martin Gander and Walter Gautschi for suggesting this composite quadrature.

The Gauss–Legendre quadrature in the interval $[-b, B]$ is then given by the nodes and weights

$$x_i = \frac{(B+b)(X_i+1)}{2} - b, \qquad w_i = \frac{B+b}{2} W_i. \tag{B.8}$$

In the interval $[-1, 1]$, the nodes $X_i$ of the Gauss–Jacobi quadrature are given by the roots of the Jacobi polynomial of degree $n$, and parameters $(\alpha, \beta)$, which we write as $J_n^{(\alpha,\beta)}$. The associated weights can be obtained from the formula

$$W_i = -\frac{2n+\alpha+\beta+2}{n+\alpha+\beta+1} \frac{\Gamma(n+\alpha+1)\Gamma(n+\beta+1)}{\Gamma(n+\alpha+\beta+1)(n+1)!} \frac{2^{\alpha+\beta}}{J_n^{(\alpha,\beta)\prime}(X_i) J_{n+1}^{(\alpha,\beta)}(X_i)}. \tag{B.9}$$

Again, we have to translate this result to the interval $(-\infty, -b]$, so we consider the change of variable $x = -b[(X+1)/2]^\beta$, with $-1 < \beta < 0$. With this change, we obtain

$$\int_{-\infty}^{-b} f(x) \mathrm{d}x = -b\beta\, 2^{-\beta} \int_{-1}^{1} \frac{f\left(-b\left(\frac{X+1}{2}\right)^\beta\right)}{(1+X)(1-X)^\alpha} (1-X)^\alpha (1+X)^\beta \mathrm{d}X$$

$$\approx \sum_{j=n+1}^{n+m} \frac{-b\beta\, 2^{-\beta} W_j}{(1+X_j)(1-X_j)^\alpha} f\left(-b\left(\frac{X_j+1}{2}\right)^\beta\right). \tag{B.10}$$

The points and weights for the Gauss–Jacobi quadrature in the interval $[-\infty, -b)$ can be read as

$$x_i = -b\left(\frac{X_i+1}{2}\right)^\beta, \qquad w_i = \frac{-b\beta\, 2^{-\beta} W_i}{(1+X_i)(1-X_i)^\alpha}. \tag{B.11}$$

The parameters $\alpha$ and $\beta$ have to be conveniently chosen so that the endpoint singularities of the integrand are absorbed into the factor $(1-X)^\alpha (1+X)^\beta$.

For the actual computation of the nodes and weights, instead of finding the roots of the associated polynomials and using equations (B.7) and (B.9), it is computationally more efficient to use the Golub–Welsch algorithm [74]. This algorithm reduces the problem to the spectral decomposition of a sparse matrix.

The kernel of the integral equation for our models is of the form

$$k(x, y) = \varphi_1(x-y) + \varphi_2(x+y), \tag{B.12}$$

where, for the $O(3)$ sigma model,

$$\varphi_1(x) = \varphi_2(x) = \mathrm{Re}\left[ \frac{\psi\left(1 + \frac{ix}{2\pi}\right) - \psi\left(\frac{1}{2} + \frac{ix}{2\pi}\right)}{4\pi^2} \right], \tag{B.13}$$

and, for Fendley's integrable models,

$$\varphi_1(x) = \mathrm{Re}\left[ \frac{\psi\left(1 + \frac{ix}{2\pi}\right) - \psi\left(\frac{1}{2} + \frac{ix}{2\pi}\right) - \psi\left(1 - \Delta + \frac{ix}{2\pi}\right) + \psi\left(\frac{1}{2} - \Delta + \frac{ix}{2\pi}\right)}{4\pi^2} \right],$$

$$\varphi_2(x) = \mathrm{Re}\left[ \frac{\psi\left(1 + \frac{ix}{2\pi}\right) - \psi\left(\frac{1}{2} + \frac{ix}{2\pi}\right) - \psi\left(\Delta + \frac{ix}{2\pi}\right) + \psi\left(\frac{1}{2} + \Delta + \frac{ix}{2\pi}\right)}{4\pi^2} \right]. \tag{B.14}$$

In both cases, the kernel $k(x, y)$ falls off as $1/x^2$ for large $x$ and fixed $y$, and analogously for large $y$, fixed $x$. This implies that the solution $v(x)$ in (B.1) also behaves as $1/x^2$ for large $x$, and thus the endpoint singularities of the integrand $f(y) = k(x, y)v(y)$ in (B.10) can be absorbed into the factor $(1-X)^\alpha (1+X)^\beta$ if we choose $\alpha = 0$, $\beta = -1/4$.

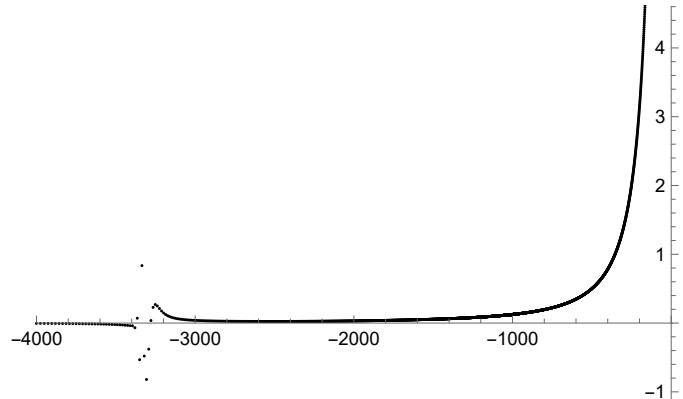

Figure 7: Numerical solution $\chi_1(\theta)$ to the integral equation (54) for the $O(3)$ sigma model at $\vartheta = \pi$, with parameters $B = 10$, $n = m = 2000$, $b = 1000$, $\alpha = 0$, $\beta = -1/4$. The solution is unreliable for large values of $|\theta|$, when $\theta < 0$.

On the other hand, we will fix the order $n$ of the Gauss–Legendre quadrature inside the interval $(-b, B]$ to be $n \approx b/2$. This gives a dense enough grid inside the interval (choosing an even denser grid does not change the numerical solution in any tangible way.)

We tested the numerical accuracy of this method with some toy models that can be solved analytically, while still retaining the main features of the actual models.[11] The general behavior is that the numerical approximations of $e$ and $\rho$ do not converge monotonically to the true result with increasing order of the quadratures $n$, $m$. Instead, the approximations oscillate around the true value. This issue can be tracked down to the numerical approximation of $v(x)$. While the approximation of $v(x)$ is in general close to the true solution for low values of $x$, it completely diverges from the true result after some point $x_0 < 0$. This behavior is illustrated in Fig. 7, for the $O(3)$ sigma model. We observed the same issue when using other quadratures, like applying the Newton–Cotes quadrature to the truncated interval $(-b, B] \subseteq (-\infty, B]$.

Therefore, we propose a different strategy to compute $e$ and $\rho$. Since $e$ is numerically better behaved than $\rho$, because the exponential weight $e^x$ in (B.2) suppresses most of the numerical errors that arise from $v(x)$ at large $x$, we focus on approximating $\rho$ and, to this end, we consider the function

$$V(x) = \int_x^B v(y)\mathrm{d}y \,. \tag{B.15}$$

Then $\rho$ can be recovered from this function as

$$\rho = \lim_{x \to -\infty} \frac{V(x)}{4\pi} \,. \tag{B.16}$$

Starting from the integral equation of (B.1), it turns out that one can find an integral equation for $V(x)$ instead, which reads

$$V(x) - \int_{-\infty}^B \left( -\int_x^B \frac{\mathrm{d}k(x,y)}{\mathrm{d}y}\mathrm{d}x \right) V(y)\mathrm{d}y = \int_x^B g(y)\mathrm{d}y \,. \tag{B.17}$$

In this case, the optimal parameters for the Gauss–Jacobi quadrature are $\alpha = 0$, $\beta = -1/3$. When numerically solving for $V(x)$, we find the same issues as with $v(x)$. Namely, the numerical solution is not reliable for large $x$. So for approximating the limit in (B.16), we will

---

[11]One obtains such a toy model, for example, by keeping only the term $\varphi_1(x - y)$ in the kernel (B.12). The resulting integral equation can then be solved exactly with conventional Wiener–Hopf techniques.

consider a large value $x_0 < 0$, which is still small enough so that numerical errors in $V(x)$ are kept under control. Then $\rho$ can be approximated as

$$\rho \approx \frac{V(x_0)}{4\pi}. \tag{B.18}$$

On the other hand, integrating by parts in (B.2), it is easy to see that $e$ can still be computed from $V(x)$ as

$$e = \frac{1}{4\pi} \int_{-\infty}^{B} e^x V(x) dx. \tag{B.19}$$

The approximation (B.18) will have two sources of errors. The first one is from an actual error in the value of $V(x_0)$, which can be improved by increasing the orders of the quadratures $n, m$. The second error comes from taking the cut-off $x_0$ instead of $x \to -\infty$. In this case, since $v(x) \sim a/x^2$ for large $x < 0$, we have

$$\left| \rho - \frac{V(x_0)}{4\pi} \right| = \left| \frac{1}{4\pi} \int_{-\infty}^{x_0} v(x) dx \right| \sim \frac{|a|}{4\pi |x_0|}. \tag{B.20}$$

This error can be improved with the Richardson extrapolation

$$V^{(1)}(x_i) = \frac{x_{i+k} V(x_{i+k}) - x_i V(x_i)}{x_{i+k} - x_i}, \tag{B.21}$$

where $k \in \mathbb{N}$. The value of $k$ can be 1, but it is better to choose a larger number to minimize error propagation when the $x_i$ are close to each other (this will always happen for very high orders $n, m$ around the point $b$). Since the subleading asymptotic correction to (B.20) is expected to involve logarithms, a second Richardson extrapolation

$$V^{(2)}(x_i) = \frac{x_{i+k}^2 V^{(1)}(x_{i+k}) - x_i^2 V^{(1)}(x_i)}{x_{i+k}^2 - x_i^2}, \tag{B.22}$$

improves the accuracy of the results (see e.g. [75, 76]). This was explicitly verified in our toy models.

In Fig. 8, we illustrate the numerical solution to the integral equation (B.17), for the $O(3)$ sigma model, and the corresponding Richardson extrapolations of (B.21) and (B.22). We see that the solution $V(x)$ and its Richardson extrapolations stabilize around a fixed value which approximates $4\pi\rho$, as predicted by (B.18). However, as $x$ increases further, the solution gives unreliable results.

For our numerical results in sections 3.3 and 4.4, we used the parameters $n = 15000$, $m = 20000$, $b = 8000$, $\alpha = 0$, $\beta = -1/3$. The weights were computed in *Mathematica* by using the Golub–Welsch algorithm, which took around one hour of CPU time. The numerical solutions were also computed with *Mathematica*, and they took around one day (for $O(3)$) or two days (for Fendley's models) of CPU time. Most of the time is spent in getting the kernel entries of (B.4), and it is strongly recommended to parallelize this part of the computation.

An interesting observation in Fig. 8 is that the first and second Richardson extrapolations have a minimum around $x \approx -4000$, instead of approaching monotonically a limit at large negative $x$. We take this behavior as an indication that after the turning point $x \approx -4000$, the numerical solution $V(x)$ is no longer reliable. Thus, we define the optimal point $x_0$ as the minimum of the second Richardson extrapolation, and take the approximation

$$\rho \approx \frac{V^{(2)}(x_0)}{4\pi}. \tag{B.23}$$

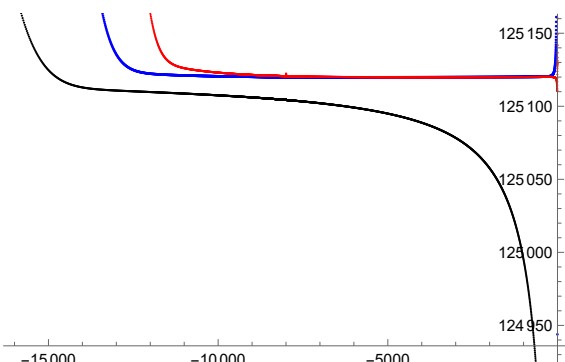

Figure 8: We plot in black the numerical solution $V(x)$ to the integral equation (B.17) for the $O(3)$ sigma model at $\vartheta = \pi$, with parameters $B = 10$, $n = 15000$, $m = 20000$, $b = 8000$, $\alpha = 0$, $\beta = -1/3$. The solution $V(x)$ stabilizes around a value, which approximates $4\pi\rho$, but then becomes unreliable for large $x < 0$. In blue and red we plot the first and second Richardson extrapolations, respectively.

We tested in the toy model that this value decreases the error by up to a factor 10 as compared to the result one would obtain when solving the original integral equation (B.1), with the same quadrature parameters.

Finally, we want to estimate the error we make when approximating the normalized energy density $e/\rho^2$. We emphasize that $\rho$ gives the main contribution to the error, and it converges very slowly as we increase the order of the quadrature. Due to both the low rate of convergence and the high computational cost of estimating this rate, a rough but accessible approximation of the error can be achieved by substantially changing the quadrature parameters. Thus, we estimate the error in $\rho$ as the difference between the approximation we obtained with the parameters $n = 15000$, $m = 20000$, $b = 8000$, and the approximation with $n = m = 3000$, $b = 1800$.

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
