# Peer review of "Instantons, renormalons and the theta angle in integrable sigma models"

_SciPost Physics, doi:SciPost Phys. 15, 184 (2023)_

## Round 2 · Referee Report · Anonymous (Referee 1) · 2023-6-4

Report

The authors compute the free energy in several series of 2d sigma models by exploiting their (conjectured) integrability. The aim is to understand the contributions of renormalons and instantons, and they do a rather intricate calcualtion to this end. While the techniques are mainly old, they successfully surmount several substantial technical obstacles to push the techniques further. In particular, they do not rely on deforming the problem, as this seems to obscure the renormalon contribution.

The question they address is interesting and the calculation meaningful. I thus recommend publication in SciPost. I do think the authors should give more explanation of a key conceptual point. I'm sure some of the below issues are addressed in the authors' other papers, but they need to say a little more here.

Namely, they need to give more detail on how renormalons appear in the the sigma model with Lie-group symmetry. but (the authors say) "do not seem to" in the deformed theory. I presume the free energy is still continuous, but is it not analytic in the deformation parameter? Or are they saying that it is not continuous?

Moreover, the old papers by Zamolodchikov et al associate the change in sign between theta=0 and pi with instantons, but at the end of section 4.3 the authors seem to be saying otherwise. How do they know this is the renormalon and not a previously missed instanton contribution?

Another issue worth mentioning is what happens in the O(3) case for varying theta continuously. Obviously, they're not going to sovle this question, but a little speculation in light of their results would be good.

Finally, a minor but useful point: the Zamolodchikov brothers adopted the convention that Alexander goes by A.B., while Alexei goes by Al. B. Most of the references here are to Alexei's papers, and so should be fixed accordingly.
  • validity: -
  • significance: -
  • originality: -
  • clarity: -
  • formatting: -
  • grammar: -

Author:  Tomas Reis  on 2023-07-21  [id 3828]

(in reply to Report 1 on 2023-06-04)

We thank the referee for their review and comments. Concerning the points raised:

1) Renormalons are non-perturbative effects defined through their manifestation in the large order behavior. Thus, they can only be identified when one has a perturbative expansion. However, in the deformed models (either O(3) or Fendley's coset sigma models), the free energy has no proper perturbative expansion. Instead, all the terms one finds are exponential corrections. In addition, even if one computes all the exponential corrections in the deformed model and then takes the deformation parameter to 0, it is not clear how one would recover the perturbative series + all exponential corrections that one finds in the undeformed model. In the revised version, we will clarify that is why we cannot have renormalons in the deformed models.

2) We do agree with the old papers by Zamolodchikov et al.: the exponential corrections that change sign in the O(3) sigma model originate from instantons. The discussion at the end of section 4.3 only concerns Fendley's sigma models introduced in section 4.1. In these models, there are both instantons and renormalons. We can (conjecturally) distinguish them by looking at the large N limit (as we explain in our Ref. [19]): instantons disappear at large N while renormalons survive. We will edit the relevant paragraph to make this discussion more clear.

3) At the end of section 3.2, in Eqs. 3.56-3.58, we speculate about generic values of the theta angle in the O(3) sigma model. The exponential correction of order exp(-2/alpha) has a real part and an ambiguous imaginary part. The imaginary part is fixed by the large order behaviour of the perturbative expansion, which we know is constant as we vary the theta angle. Therefore, only the real part of the exponential correction might change. This led us to speculate that the real part of the exponential correction should be multiplied by cos(theta), as is expected from instantons, which gives Eq. 3.56. We finish with a concrete prediction for the topological susceptibility in 3.58.

4) We will update the references according to the convention and we thank the referee for pointing it out.

---

## Round 2 · Referee Report · Anonymous (Referee 2) · 2023-7-17

Report

In this work the authors exploit Bethe ansatz methods to compute the free energy in various integrable $2$d qfts. In particular, they find transseries expansions for the free energy of the $O(3)$ (and other integrable coset models) at $\theta=0$ and $\theta = \pi$.

Most of the analysis closely follows the methods introduced in a previous work by the same authors, while the novelty here regards the change in NP corrections as the theta angles varies between $0$ and $\pi$.

Although the key discussions in section 3 and 4 are rather technical, I found the paper to be clear and well-written, certainly deserving publication with Scipost.

My main issue with the paper lies with the use of the nomenclature "renormalons", to describe certain non-perturbative corrections found in the weak-coupling regime.

The present results are derived via Bethe ansatz equations and do not rely on any diagrammatica of sort. Although I do agree with the authors that such corrections are "renormalons" in nature, when compared to other "instanton" corrections, in that they survive in the large-$N$ limit, this naming seems very misleading.

I think a better way of addressing the problem is the following question: do these "renormalons" non-perturbative terms the authors have found come from finite-action semi-classical solutions to the (possibly complexified) classical field equations of the $O(3)$ sigma model in a magnetic field background?

In the authors calculation, the magnetic field, h, plays a crucial role in establishing the weak-coupling expansion, i.e. $h/m \to \infty$ with $m$ the mass-gap. In the very same limit, $ h\to \infty$, the O(3) sigma-model path-integral, with or without theta-angle, should be amenable to a reliable semi-classical expansion.

Can the authors find semi-classical solutions to the complexified field equations and responsible for these "renormalons" non-perturbative terms? Alternatively, do the authors have any argument, besides large-$N$, for why there should not be any semi-classical object responsible for such corrections?

I would like the authors to comment on this.
  • validity: -
  • significance: -
  • originality: -
  • clarity: -
  • formatting: -
  • grammar: -

Author:  Tomas Reis  on 2023-07-21  [id 3829]

(in reply to Report 2 on 2023-07-17)
Category:
answer to question

We would like to thank the referee for the comments.

Indeed, the identification of exponentially small corrections as coming from “instantons” or “renormalons” is not an exact science. Yet. In our paper we have used the rule of thumb that, at infinite volume, exponentially small corrections which are not suppressed in the large N limit are unlikely to be due to classical configurations. Since at large N the number of diagrams only grows exponentially, these corrections lead to a factorial large order behavior in perturbation theory which must be due to integration over momenta. Therefore, the name “renormalons” seems to be justified in this sense.

The referee points out that, since we are working at weak coupling, our calculation "should be amenable to a reliable semi-classical expansion.” However, there is substantial evidence from renormalon physics that this statement might not be generically true, since in generic renormalizable theories, exponentially small corrections arise whose origin does not seem to be semiclassical, and might involve intrinsic non-perturbative physics of a different type (some people have suggested that they might involve vacuum condensates of operators, which are definitely not semiclassical). Let us point out that, although renormalons were found almost 50 years ago, to our knowledge nobody has ever identified a renormalon correction as due to a semi-classical configuration by a first principles calculation (certainly not in infinite volume).

Can the authors find semi-classical solutions to the complexified field equations and responsible for these "renormalons" non-perturbative terms? Alternatively, do the authors have any argument, besides large-N, for why there should not be any semi-classical object responsible for such corrections?

Conventional solutions to the EOM with finite action in infinite volume do not seem to explain these configurations. Although we don’t have any proof that such configurations do not exist in a suitable complexified version of the theory, we believe that the burden of the proof should rather be on the people who claim that there is such a semiclassical origin. We remain agnostic on the issue, and in the absence of a positive proof of such an origin, we feel that calling these exponentially small corrections “renormalons” is rather reasonable.

---

## Round 3 · Referee Report · Anonymous (Referee 1) · 2023-9-10

Report

The authors addressed my comments in my first report, and so I recommend publication in SciPost in this form.

---

## Round 3 · Referee Report · Anonymous (Referee 2) · 2023-9-25

Report

The authors have scantly addressed the comments/questions I raised. I can however recommend publication in SciPost.

---

## Round 3 · List of Changes

1) We clarified the discussion of the deformed "sausage" models to better explain why there are no renormalons in such models. This affects the relevant paragraph in section 1.

2) We added details to the discussion of the non-perturbative effects in the coset sigma models to clarify why some are classified as instantons and other as renormalons, as well as the implications of such distinction. This affects the end of section 4.2 and a little bit of section 5.

3) We updated the references to the convention where Alexei Zamolodchikov is referred as Al. B. Zamolodchikov while Alexander Zamolodchikov is referred as A. B. Zamolodchikov.

---

## Editorial Decision

published